# Oxide interface-based polymorphic electronic devices for neuromorphic computing

Soumen Pradhan [1] ✉, Kirill Miller[1], Fabian Hartmann [1] ✉, Merit Spring[1], Judith Gabel[1], Berengar Leikert[1], Silke Kuhn[1], Martin Kamp[1], Victor Lopez-Richard [2], Michael Sing [1], Ralph Claessen [1] & Sven Höfling[1]

Aside from recent advances in artificial intelligence (AI) models, specialized AI hardware is crucial for addressing large volumes of unstructured and dynamic data. Conventional complementary metal-oxide-semiconductor (CMOS)-based AI hardware faces several critical challenges including scaling limitations, the separation of computation and memory units, and overall system energy efficiency. While emerging materials have been proposed to overcome these limitations, issues such as scalability, reproducibility, and compatibility remain critical obstacles. Here, we demonstrate polymorphic electronic devices with programmable transistor, memristor, and memcapacitor functionalities by manipulating the quasi-two-dimensional electron gas in $LaAlO_3$/$SrTiO_3$ heterostructures using lateral gates. A circuit utilizing transistor and memcapacitor functionalities exhibits digit recognition, enabling implementation in physical reservoir computing. An integrated circuit incorporating transistor and memristor functionalities performs logic operations with in-situ output storage and supports advanced reconfigurable synaptic logic operations for multi-input decision-making tasks such as patient monitoring. Our findings pave the way for oxide-based monolithic integrated circuits in a scalable, silicon-compatible, energy-efficient single platform for polymorphic and neuromorphic computing.

In the era of advanced artificial intelligence (AI) technology, the increasing volumes of data generated and updated in our daily lives have created high demand for fast and energy-efficient computing systems[1]. However, training of AI models requires enormous amounts of energy[2,3], especially when implemented in traditional complementary metal-oxide-semiconductor (CMOS) technology-based electronic hardware, due to limitations in device scaling[4,5]. As a promising alternative, polymorphic technology—a special class of reconfigurable technologies—is capable of reconfiguring its hardware functionality irrespective of time and space[6], simplifying circuitry by

reducing the number of electronic components, thereby decreasing both the area and energy consumptions[7]. On the other hand, to overcome the von Neumann bottleneck[8], human brain-inspired neuromorphic computing has gained significant attention as a new computing paradigm that offers parallel signal processing with low energy consumption[8,9]. Reservoir computing (RC), as one of them, needs training only on the 'readout function' to produce a desired output, which significantly reduces the training costs and can be employed in state-of-the-art hardware prototypes for pattern recognition[10], signal processing in noisy environments[11] and unsupervised learning[12].

[1]Julius-Maximilians-Universität Würzburg, Physikalisches Institut and Würzburg-Dresden Cluster of Excellence ctd.qmat, Am Hubland, Würzburg, Bavaria, Germany. [2]Universidade Federal de São Carlos, Departamento de Física, São Carlos, SP, Brazil. ✉e-mail: soumen.pradhan@uni-wuerzburg.de; fabian.hartmann@uni-wuerzburg.de

Notably, neuromorphic transistors have the ability to reconfigure logic operations[13] and store the logic output[13,14], enabling their decision-making capabilities[15] and applications in adaptive learning[16], and edge computing[17].

Till today, extensive research has been conducted on neuromorphic computing based on a wide range of material systems, including two-dimensional (2D) materials[18,19], inorganic compounds[20], organic materials[21] and oxides[22]. Among them, 2D-materials-based neuromorphic devices have attracted significant attention due to their exceptional electronic and optical properties[23–25]. Additionally, 2D materials have emerged as promising candidates for reconfigurable technologies[26,27], alongside silicon nanowires[28]. However, the structural complexity of most of the reported devices raises the manufacturing costs. Also, these devices face challenges, including wafer-scale fabrication, integration with existing technologies, performance variability, and most importantly, degradation in air, which impacts long term stability[29,30]. In contrast, oxide materials continue to garner interest as they effectively mitigate many of these limitations, offering enhanced durability, scalability and compatibility with conventional fabrication processes[31–33].

In this context, the discovery of a highly mobile quasi-two-dimensional electron gas (q2-DEG) at the interface of LaAlO$_3$/SrTiO$_3$ (LAO/STO) heterostructures has paved the way for the development of oxide-based next-generation electronic devices[34,35]. However, the manipulation of the q2-DEG, using metallic top-gate electrodes, results in additional band bending at the interface[36–38], while back-gate electrodes are limited to global control of the q2-DEG and often require cryogenic temperatures[39]. Hence, a viable and better alternative is laterally defined side-gates alongside a nanowire channel, which can be processed easily on the surface and have shown considerable potential in LAO/STO heterostructures[40–43]. Beyond conventional field effect functionalities, employing side-gates also leads to the emergence of memristive operation, attributed to the migration of oxygen vacancies or floating gate effects[41,44]. Interestingly, the memristive properties of these devices can be intrinsically linked to memcapacitive behavior[45]. However, only a few studies have reported hysteresis in capacitance in LAO/STO heterostructures ascribed to oxygen vacancies migration[46], structural distortion[36] or interfacial trap state[47].

In this study, we present a polymorphic electronic device that leverages the q2-DEG at the LAO/STO interface, capable of operating as a transistor (T), memristor (M), or memcapacitor (MC) at room temperature, depending on the biasing condition. By integrating 1T with 1MC, an RC system is implemented using a 4-bit pulse scheme. Furthermore, combining 2T and 1M, logic OR and AND operations are demonstrated, with the significant advantage of in-situ data storage within the computing circuit. Notably, both logic functions are reconfigurable within a single circuit design, allowing their implementation in programmable operating systems. These versatile functionalities, along with the broad range of combinatorial applications, highlight the device's potential for further advancing the field of oxide electronics.

## Results
### Polymorphic electronic devices
The LAO/STO-based nanoelectronic devices composed of a nanowire channel and two lateral gates were fabricated in only three main steps. At first, a TiO$_2$-terminated (001)-oriented STO substrate was patterned by electron-beam lithography after spin-coating a negative photoresist. An 11 nm SiO$_2$ layer was then deposited via electron-beam evaporation, and a lift-off process was used to define the nanowire and two rectangular gate regions at both sides to ensure direct access of STO surface on these selected areas. Finally, a 6-unit-cell LAO film was grown by pulsed laser deposition. LAO deposited directly on STO crystallized (termed as 'c-LAO') and formed the q2-DEG at the interface, whereas LAO on SiO$_2$ remained amorphous (termed as 'a-LAO')

and insulating. A detailed description of the fabrication process is included in the "Method" section. An atomic force microscopy (AFM) image of a single device with notations of the structural hierarchy of the layers is shown in the top left panel of Fig. 1a. The AFM image confirms the well-defined structure of the device with a nanowire conducting channel and two lateral gates, where LAO was directly grown on TiO$_2$-terminated STO substrate to form the q2-DEG. The mobility of the q2-DEG formed at the LAO/STO interface in our devices is approximately 5.44 cm$^2$/(Vs) at room temperature. Its temperature variability was reported in our old publication[44]. Now, we focus on the polymorphism of a single device based on different wiring configurations. The schematic illustration for field effect transistor (FET), memristive, and memcapacitive functionalities are depicted in Fig. 1a with a corresponding circuit diagram alongside the AFM image of the device. However, to examine the FET characteristics of the device, the gate voltages are applied to the lateral gates as shown schematically. The left panel of Fig. 1b shows the output characteristics for varying gate voltage ($V_G$). As drain voltage ($V_D$) increases, the drain current ($I_D$) exhibits saturation behavior similar to an n-channel FET. The transfer characteristics (right panel of Fig. 1b) quantify the device's response to varying gate voltage at different $V_D$ values. The results demonstrate an on-current in the $\mu$A range with an on/off current ratio in the range of ~10$^5$, confirming robust gate modulation. Notably, the robustness of gating functionality over time strongly depends on the stability of q2-DEG. Previous studies showed that even for crystalline LAO films, interfacial atomic interdiffusion or local chemical reactions can occur during growth, but are typically limited to the very first atomic layer of LAO[48]. Such limited intermixing does not suppress the polar catastrophe-driven electronic reconstruction responsible for q2-DEG formation. Moreover, a 6 u.c thick crystalline LAO layer exhibits negligible time-dependent variation of sheet resistance, in contrast to thinner crystalline films (≤4 u.c) or amorphous LAO, grown on STO, as reported by Trier et al.[49]. Therefore, we can confirm the long-term stability and reliability of both the conduction channel and gating effect.

To switch the device functionality to memristor, the full cycle $I_D$ ($V_D$)-characteristics are examined for grounded and floating gates configurations, showing their dependency on gate potential (Fig. 1c). When the side-gates are grounded, the $I_D$ ($V_D$)-curve exhibits no memory effect and no hysteresis. In contrast, with the gates floating, a pinched hysteresis loop emerges with two distinct resistance states near zero bias voltage, indicative of memristive response. The gate potential dependence of $I_D$ ($V_D$)-characteristics rules out the oxygen vacancy migration as the main mechanism of memristive response. In contrast, the memory effect arises from the charging and discharging of the lateral gates, which occurs via charge tunneling through the gate dielectric between the gates and the conducting channel. In the case of grounded lateral-gates, any charge transferred to the gates is drained, preventing charge accumulation and, subsequently, the development of hysteresis. Conversely, in the floating gate configuration, the side gates accumulate or release charges gradually, depending on drain voltage, which ensures the hysteresis in the full cycle map for an adequate voltage sweeping rate[41]. This process can be quantitatively described by the theoretical framework introduced in our Ref. 50, which couples diffusive transport along the two-dimensional conductive channel with capacitive coupling between the channel and the gates. The simulated output characteristics of the device at stable conditions under grounded and floating gate conditions are found to be similar to the experimental data (Supplementary Fig. 1a). Here, the time-dependent charging and discharging of the floating gates follow the generating function illustrated in Supplementary Fig. 1b, which describes electron discharging at positive bias and charge trapping at negative bias. The specific nature and forms of these generating functions, which determine the temporal response of the floating gates, are further discussed in Refs. 51 and [52]. Anyway, two resistance

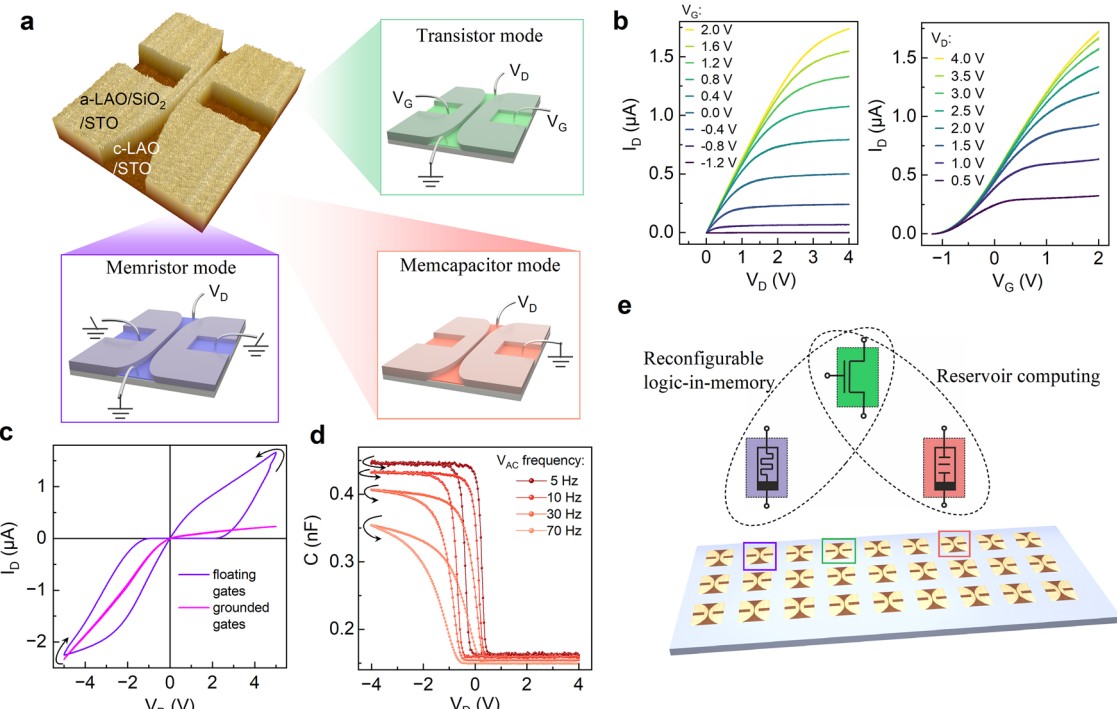

**Fig. 1 | Polymorphic functionality of a runtime reconfigurable oxide interface-based device. a** Atomic force microscopy image of the device with a scanning area of 5 × 5 μm² alongside the schematic illustrations with circuit diagram for transistor, memristor, and memcapacitor operation modes. **b** Output characteristics of the device for different gate voltages, $V_G$ (left panel), and transfer characteristics for different drain voltages, $V_D$ (right panel). The device operates as a planar n-channel field-effect transistor. The channel can be depleted depending on the gate potential. **c** $I_D$ ($V_D$)-characteristics for floating and grounded gates. The device can be

operated as a memristor (type-I) or a non-linear resistor controlled by the gate potential. **d** Capacitance-voltage-characteristics measured between the wire and one of the gates, while the other gate is left floating. The arrows indicate the direction of the sweep cycles. **e** Schematic illustration showing reservoir computing applications integrating transistor-memcapacitor devices and reconfigurable logic-in-memory applications in a transistor-memristor integrated circuit in LAO/STO-interface-based single platform.

values of 8.52 GΩ ($R_{high}$) and 1.53 MΩ ($R_{low}$) extracted from the linear fits of the experimental forward and reverse bias data near zero bias yield a resistance ratio of approximately $5.6 \times 10^3$ for the device. The result corresponds to a type-I memristive response, which, according to Ref. [53], is expected to arise under polarity-dependent charge accumulation or release. When a symmetric voltage sweep is applied with respect to the middle of the nanowire, a type-II memristive response can be observed (Supplementary Note 1, Supplementary Fig. 2).

The third functionality of the device emerges from its memcapacitive response with hysteresis in capacitance-voltage-characteristics along with a distinct transition between high and low capacitance states (Fig. 1d). The capacitive response observed here differs from previous reports that attributes to structural distortion[36], large geometric capacitance[54], or oxygen vacancy migration[46]. In this device, the layout and the dynamics of charge trapping and detrapping provide the explanation for the capacitive transitions. Here, the device operates similarly to a metal-oxide-semiconductor diode, where the capacitance exhibits an asymmetric response (Supplementary Fig. 3). Under accumulation conditions (reverse bias), the capacitance saturates at its higher value, corresponding to $C_{high} \sim 1/d_B$, where $d_B$ is the distance between the channel and the gate. Under depletion conditions, the capacitance decreases to $C_{low} \sim 1/(d_B + d_D)$, where $d_D$ represents the depletion length. A comprehensive theoretical framework capturing the capacitance modulation through charge localization dynamics on the floating gate, consistent with the experimental C−V characteristics, is presented in more detail in Ref. [55].

Consequently, transistor, memristor and memcapacitor functionalities can be programmed in a single device depending on the

circuit arrangement. Implementing these functionalities using separate devices typically requires a footprint of 4–15 μm² when including the transistor selector (1–3 μm²), a memristor element (1–2 μm²), a capacitor stack (1–3 μm²), and interconnect overhead (1–3 μm²). Our LAO/STO lateral polymorphic device occupies only ~1 μm², resulting in a 4–10 times reduction in area. Moreover, since all programmable operations are enabled by the polymorphism of the presented device, an architecture with multiple functionalities avoids, e.g., driving multiple interconnects, charging/discharging separate capacitive nodes, losses in selector transistors, etc. The realization of a single device platform will thus provide an efficiency advantage on the integrated-circuit (IC) level rather than on the device level itself. Most importantly, eliminating two additional devices reduces interconnect routing by 50–70% and improves projected per-tile yield by 30–40%. Therefore, our LAO/STO interface-based polymorphic devices are advantageous regarding footprint, energy consumption, and projected per-tile yield compared to separate devices. Here, transistor functionality results from the coulomb interactions between q2-DEG on the lateral gates and channel. On the other hand, the memristive and memcapacitive functionalities originate from the same mechanism of charge trapping/detrapping on the lateral floating gate. Therefore, it becomes necessary to check if there is any interference of one functionality into another. Thus, we perform the cycle measurements on a single device in transistor, memristive and memcapacitive configurations, consecutively (Supplementary Note 2, Supplementary Figs. 4–6) and confirm the cyclic endurance of all modes. In the memristive configuration, consecutive cycling measurements reveal a small initial transient in the $R_{off}/R_{on}$ ratio around the $10^3$ range, followed by an approximately linear increase with an average increment rate of 0.27

per cycle and a standard deviation of 103 extracted from the regular residuals. In the memcapacitive configuration, repeated cycling yields well-defined high and low capacitance states of 432.0 ± 0.8 pF and 159.0 ± 0.7 pF, respectively, at zero bias. Most importantly, cycling of one mode does not degrade another mode's operation. Moreover, the long-term stability of the device operation was confirmed from the years of operational functionality (Supplementary Note 3, Supplementary Fig. 7). Now, we emphasize the strength of our polymorphic oxide devices for application-specific adaptability. Peng Wu et al. reported that black phosphorus-based reconfigurable FETs can be utilized in security cells incorporating polymorphic NAND/NOR logic gates, advancing their potential for hardware security applications[56]. In another report, a scalable single-gate transistor using sub-stoichiometric zirconium oxide and molybdenum disulfide with reconfigurability between transistor and diode demonstrates both the photo-switching and photo-synaptic functionalities[57]. Here, building on the primary functionalities, new integrated operations such as RC and reconfigurable logic-in-memory architectures are explored as schematically represented in Fig. 1e, which will be discussed in the following sections.

## Reservoir computing

First, we demonstrate a neuromorphic computing architecture for RC application by connecting one transistor and one memcapacitor (Fig. 2a). Notably, compared to transistor- and memristor-based RC systems where the reservoir state is defined by output current, memcapacitor-based devices offer key advantages: the voltage output eliminates static current induced power dissipation and the need

for current-to-voltage conversion[58]. To investigate the RC operation, an input voltage pulse ($V_G = 3$ V) is applied to turn 'on' the transistor for different $V_D$ values. As depicted in Fig. 2b, for $V_D$ of 1 and 2 V, the output voltage ($V_O$) reaches $V_D$, while in the case of 3 and 4 V it reaches 2.47 V and 2.57 V, respectively, which occurs due to the polarity inversion between the drain and gate (explained in Supplementary Note 4, Supplementary Fig. 8). Furthermore, $V_O$ does not remain constant over time-even after the transistor is turned 'off'-instead $V_O$ decays gradually, indicating charge leakage from the memcapacitor. This leakage can be controlled by increasing the separation between the gate and channel, and also by replacing $SiO_2$ with a high-$\kappa$ dielectric material as the gate oxide layer. Anyway, the decay in $V_O$ in the present device confirms the presence of short-term memory with the memory window extending as $V_D$ increases. Now, keeping $V_D$ fixed at 4 V, we vary the input pulse width and show a representative time window in Fig. 2c (full range in Supplementary Fig. 9). The results reveal a nonlinear increase of $V_O$ with prolonged memory retention as we increase the pulse width. The quantification of the temporal memory is detailed in Supplementary Note 5, Supplementary Fig. 10. Interestingly, the non-linearity and temporal memory are the two main criteria for hardware RC systems, where the temporal memory ensures that the reservoir state encodes a decaying trace of recent input history, providing access to multiple temporal delays. At the same time, nonlinearity transforms these historical inputs into a high-dimensional and linearly separable representation. The combination enables a simple linear readout to solve nonlinear and temporally complex tasks—something impossible for a purely linear or memoryless system. Without fading

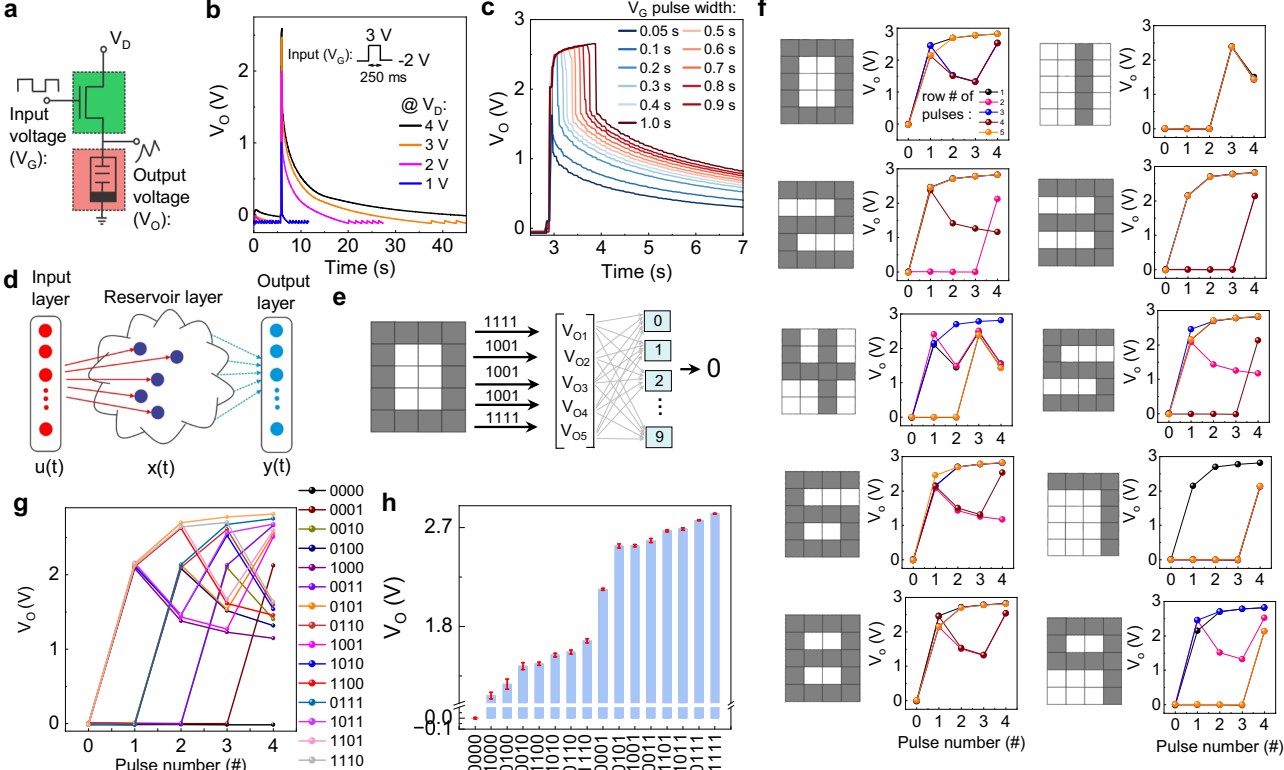

**Fig. 2 | Reservoir computing operation in an integrated circuit with one transistor and one memcapacitor. a** Schematic illustration of a 1T1MC integrated circuit with input voltage pulse to the gate ($V_G$) of T and output voltage ($V_O$) from MC while a drain voltage ($V_D$) is applied to the drain terminal of T; synaptic $V_O$ when applying a single input pulse between −2 and 3 V with **b** a fixed pulse width of 250 ms and varying $V_D$, and **c** varying pulse widths from 0.05 s to 1 s and fixed $V_D$ of 4 V; **d** schematic representation of a reservoir computing (RC) system with the reservoir layer directly connected to the output layer and only the output layer being trained to build the weight matrix, **e** experimental demonstration of pattern recognition of a 0 digit image of 5 × 4 pixels with an array of 5-reservoir outputs for pattern classification, **f** $V_O$ for each pulse stream corresponding to each row in the digit images from 0 to 9, **g** $V_O$ for all 16-types of 4-bit pulse trains and **h** average $V_O$ with error bars at the end of all 16 states considering cycle to cycle variation.

memory, the reservoir cannot exploit temporal correlations, while long-term memory remembers not only the recent past input but also the input applied to the distant past, thereby making it difficult for distinguishable output[59]. On the other hand, without nonlinearity, the reservoir collapses into a linear filter incapable of computing linearly inseparable tasks (detailed in Supplementary Note 6). Therefore, our device configuration is well-suited for physical RC applications, serving as an effective reservoir layer that fulfills both the non-linearity and temporal memory requirements.

A physical RC system comprises of three layers, where the input layer is mapped to high-dimensional space through the reservoir layer, which directly feeds into the output layer for direct classification (Fig. 2d). To demonstrate this concept, consider a monochrome image of n × m pixels, where each row of pixels is input sequentially to an array of n 1T1MC devices, which acts as the reservoir layer. The output voltages from each device are fed into the classification network in the output layer, consisting of $d$-output neurons, where $d$ represents the number of classification labels. For classification, the dot products of the reservoir output, represented as an n × 1 vector, are computed with the n × d weight matrix. The neuron label corresponding to the maximum dot product is identified as the predicted final output. The output layer requires supervised training to optimize the weight matrix for accurate classification. To illustrate this approach, we perform a digit recognition task using computer-generated 5 × 4 digit pixel images from 0 to 9. Figure 2e illustrates the schematic for digit 0 as an example. Figure 2f displays the digit images alongside the output voltages corresponding to each 4-pixel pulse stream. As observed, $V_O$ is progressing at each pulse "1", while decaying towards its initial state with each "0" pulse, resulting in distinct $V_O$ values at the end of each pulse train based on the input sequence. The collective reservoir state

forms a unique 5 × 1 matrix for each digit. The reservoir state can then be effectively classified in the output layer through training. Note that only 6 out of 16 possible 4-bit combinations are used to represent the monochrome image of 10 digits. To further explore the device response, all 16 possible 4-bit pulse trains were applied to the device from its resting state (Fig. 2g). To assess the consistency of the device response, each 4-bit pulse train was cycled 10 times. The average $V_O$ after stimulation, along with error bars, indicates that almost all configurations can be reliably distinguished (Fig. 2h). The contrast among the outputs can be improved by tuning the parameters such as pulse width, inter-pulse gap and pulse amplitude. Now, to compare the output with the linear system, similar pulsing schemes are applied to our device in transistor mode in series with a commercial capacitor. The output shows independence with the arrangement of pulses, resulting in failing to demonstrate meaningful RC operation (Supplementary Note 6, Supplementary Fig. 11). Therefore, our devices can be utilized in hardware-based RC systems as a reservoir layer for pattern recognition. Moreover, an energy analysis shows that the total energy drawn per pulse by our 1T1MC-based RC system is ~ 2.71 nJ. In contrast, a 1T1M-based RC system would consume ~ 16.57 nJ under matched operating points. A detailed energy analysis, including pulse energy, energy per inference and average power for our 1T1MC-based RC system is shown in Supplementary Note 7.

## Synaptic plasticity

Artificial neural networks based on memristors are typically arranged in a crossbar layout. To address individual synaptic nodes and minimize sneak current paths into neighboring cells, a 1T1M configuration is employed, where the transistor acts as a selector[60] (Fig. 3a), leveraging the device's inherent reconfigurability without compatibility

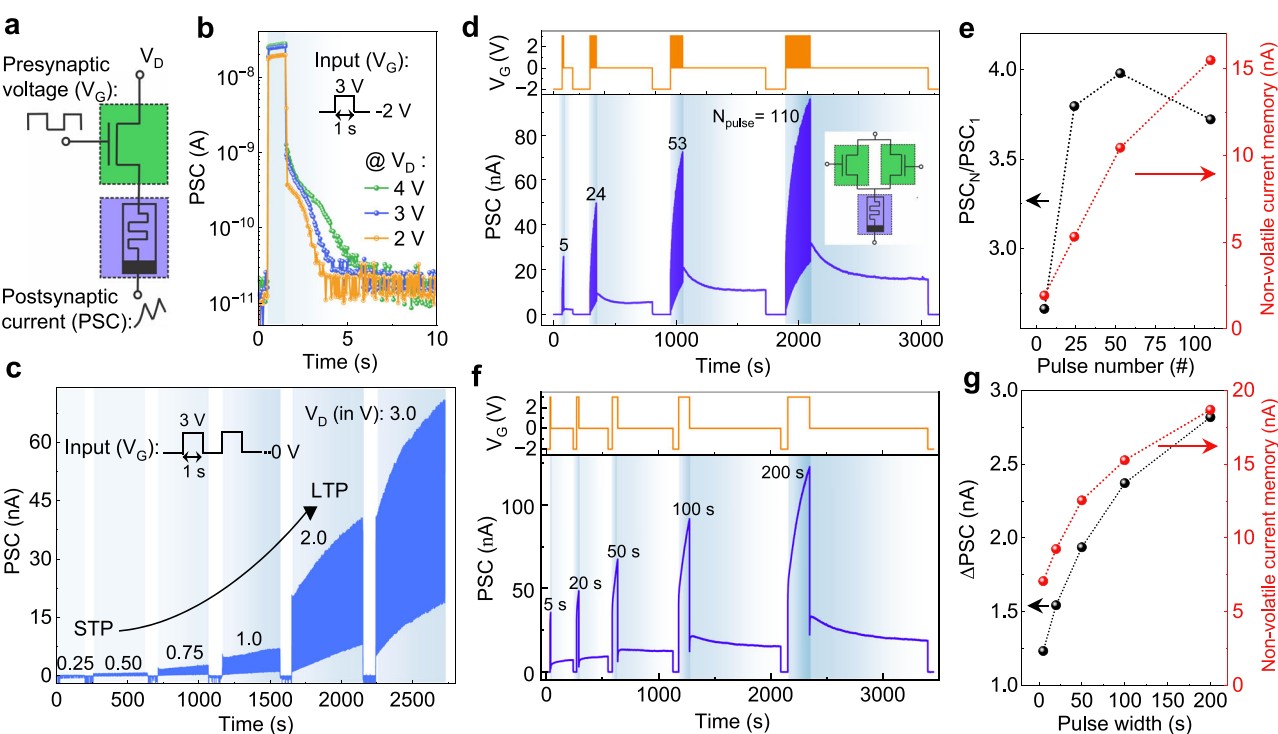

**Fig. 3 | Short and long-term synaptic operation by integrating transistor and memristor devices. a** schematic diagram of a 1T1M combined circuit with input voltage pulse to the gate ($V_G$) of T and postsynaptic current (PSC) from M while the voltage ($V_D$) applied to the drain of T is read, **b** short term potentiation (STP) in the circuit by switching `on' the transistor with a single $V_G$ pulse between −2 and 3 V at different $V_D$, **c** post synaptic current (PSC) for continuous input voltage pulse train of 3 V with different $V_D$ from 0.25 V to 3 V, indicating the transition from STP to long term potentiation (LTP) with increasing $V_D$. Transition from STP to LTP in a 2T1M

integrated circuit as seen in the PSC by changing **d** the $V_G$ pulse number, **f** the $V_G$ pulse width for $V_D$ = 4 V. Inset in **d** shows the schematic diagram of a 2T1M combined circuit with input $V_G$ pulse to the gates of both T and PSC from M. Spike number and width-dependent plasticity by showing **e** variation of $PSC_N/PSC_1$ and non-volatile memory with pulse number, **g** variation of ΔPSC and non-volatile memory with pulse width indicating higher non-volatile memory for longer pulse width and higher pulse number.

issues. To investigate memory characteristics, a single $V_G$ pulse is applied to switch the transistor 'on' for different $V_D$ values (Fig. 3b). Here, the post-synaptic current (PSC) rapidly reaches its maximum value, then retains an intermediate value for a few seconds after the transistor is switched 'off' before gradually returning to the baseline 'off' current. This transient retention indicates short-term potentiation (STP) within the device. Furthermore, Fig. 3c displays PSC as a function of time under a train of $V_G$ pulses of amplitude 3 V for various $V_D$ values. At lower $V_D$, PSC remains unchanged despite successive pulses. However, at higher $V_D$, PSC progressively increases with each subsequent pulse, demonstrating a transition from STP to long-term potentiation (LTP). Additionally, the system can be reset to its initial state by fully turning 'off' the transistor before proceeding to the next voltage condition.

In the biological brain, continuous or repetitive stimulation of signals strengthens the synaptic weights, enabling the storage of information over extended periods, facilitating the transition from STP to LTP. To mimic this operation in our device, synaptic properties are extracted by varying input pulse width and count for the 1T1M device configuration (Supplementary Note 9, Supplementary Fig. 13). For the 2T1M circuit configuration, as schematically illustrated in the inset of Fig. 3d, we performed similar experiments by applying identical $V_G$ pulses to the gates of both transistors. Figure 3d, f depict the PSC as a function of time together with varying input pulse number and width, respectively, at $V_D = 4$ V. The PSC increases non-linearly with pulse count and width, which can be attributed to the cumulative discharge from the floating gates to the nanowire. After each pulse scheme, the PSC initially decays exponentially before gradually saturating over time, indicating non-volatile memory characteristics. For better visualization of these results, we extracted the non-volatile current memory, spike-number-dependent-plasticity ($PSC_N/PSC_1$) and spike-width-dependent-plasticity ($\Delta PSC$) (Fig. 3e, g). Initially, $PSC_N/PSC_1$ increases non-linearly with the number of pulses, but beyond 110 pulses, the ratio begins to decrease. This suggests that after a certain threshold, the charging rate on the floating gate increases compared to the discharging rate. However, $\Delta PSC$ and the overall non-volatile memory increase with the number and width of pulses, confirming the transition from STP to LTP. This behavior demonstrates that our devices can efficiently emulate key functions of neuromorphic technology, enabling learning, temporary forgetting, relearning and long-term information retention.

## Logic computation and its memory

In logic computation OR, AND and NOT gates are fundamental building blocks, as all other logic gates can be constructed from them. NOT gate operation is demonstrated utilizing one T and one resistor (Supplementary Note 10, Supplementary Fig. 14). Here, we demonstrate OR and AND logic operations using a 2T1M configuration. To perform logic operations, the transistors' gate voltages (denoted as $V_{G1}$ and $V_{G2}$) serve as input signals, while the current ($I_{out}$) through the memristor represents the output signal. Specifically, a negative voltage (−2 V) is defined as logic input "0", and a positive voltage (3 V) as logic input "1". The $I_{out}$ of 4 nA is considered as the threshold between the logic output "0" and "1". For the OR gate operation, corresponding to the circuit shown in the inset of Fig. 4a, when both the inputs are logic "0", the $I_{out}$ is in the pA range, representing output "0" (Fig. 4a). For all other input combinations, a significantly higher $I_{out}$ than 4 nA is observed, corresponding to logic output "1". This results in a clear distinction by 3 orders of magnitude between the "low" and "high" logic states, confirming the successful OR gate operation. Similarly, AND gate functionality is validated using the circuit shown in the inset of Fig. 4b. Consistent with the OR gate operation, $I_{out}$ of few tens of pA and nA are observed for "00" and "11" inputs, respectively (Fig. 4b). For logic inputs "10" and "01", $I_{out}$ increases by one order of magnitude compared to "00", but still remain below the threshold value. Therefore, the logic operation illustrated in Fig. 4b demonstrates the functionality of an AND gate. To verify the consistency and reliability of the logic performance, all possible sequences of logic inputs are applied to both the OR and AND gate configurations.

As previously discussed in Fig. 3, the 2T1M circuit configuration can also support long-term memory through prolonged input stimulation. Here, the logic output memory is examined by monitoring the $I_{out}$ after setting the inputs to 0 V, following their stimulation for 100 s for both the OR (Fig. 4c) and AND (Fig. 4d) logic. In all cases, immediately after input removal, there is a sudden fluctuation in $I_{out}$. However, in the case of output memory for logic input "00" for OR gate, $I_{out}$ gradually increases with time and crosses the threshold current level of 4 nA after approximately 370 s as shown in the left panel of Fig. 4c. Therefore, we can say that the logic output, "0" is maintained for that time for the "00" input condition. To examine the reset latency, both inputs were again set to −2 V and an immediate drop in $I_{out}$ to the pA range is observed. However, for the same device configuration, during the examination of logic-output memory for the input conditions: "10", "01", and "11", $I_{out}$ decreases slowly after an immediate drop and then stabilizes at a value well above 4 nA, even after 1500–2000 s, confirming storage of logic output, "1" for a long duration. These results confirm that all the "high" logic outputs can be stored for an extended period in the circuit, compared to the "low" logic output for the OR gate. Similarly, during the logic memory test for the AND-gate operation for the logic inputs: "00", "10", and "01", $I_{out}$ increases slowly after an immediate jump, and then saturates at a value well below the 4 nA threshold during the memory test of approximately 1500 s. In contrast, for the logic input "11", $I_{out}$ remains above 4 nA for the entire 1500-s duration after initial fast decay and eventually saturates over time. Therefore, the logic output is stored for a longer time period in the circuit corresponding to the AND gate. The underlying mechanism of logic memory retention is detailed in Supplementary Note 11, Supplementary Fig. 15. The insets in the right panels of Fig. 4c, d present the corresponding truth tables for both the OR and AND gates, summarizing the logic outputs during active input processing and their memory states. During the memory test, once saturation in current is observed, the gate voltages were reset to −2 V and a sudden drop in currents was recorded for the reset latency test. The results were analyzed in detail in the Supplementary Note 12, Supplementary Fig. 16. In addition, the cycling endurance test of logic operations was also performed to investigate the non-volatile behavior and operation limits, including any drift in the logic operations (Supplementary Note 13, Supplementary Fig. 17), and the results confirm the robustness of the logic operations. Therefore, non-volatility of logic operations with very low reset latency and in-situ logic output storage for an extended period for OR and AND gates, in addition to NOT gate operation in the system, displays its potential in a universal set of logic gate operations.

## Reconfigurable synaptic logic

To mimic the decision-making complex functionalities of the biological brain, one requires the ability to dynamically reconfigure logic computations based on external conditions. Here, the reconfigurability of logic operations is examined via the phase of the $V_D$ sweep direction for the 2T1M circuit shown in Fig. 4a (Supplementary Note 14, Supplementary Fig. 18). We recorded $I_{out}$ over time at $V_D = 3$ V for all input combinations, following $V_D$ sweeps from −4 to 3 V and from 4 to 3 V for the AND and OR logic operations, respectively (Fig. 5a, b). It confirms that the phase of the $V_D$ sweep cycle effectively adds a new control dimension for logic computation and opens new possibilities for decision-making applications. For instance, these devices can be integrated into bioelectronic platforms to support diagnostic tasks[15]. Here, we design a simple diagnostic model to monitor two types of person with preconditions, healthy or heart-disease and two

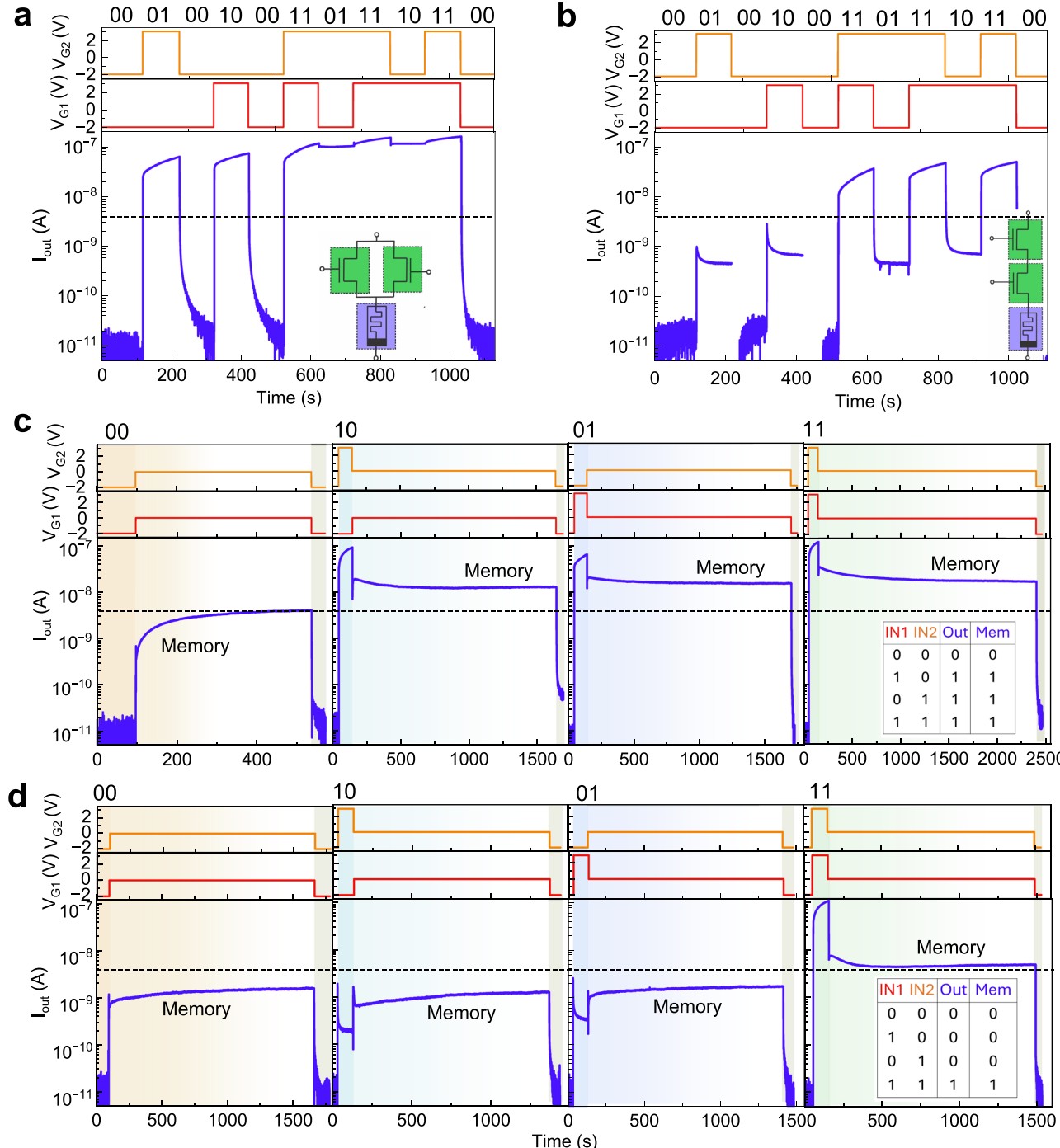

**Fig. 4 | Logic OR and AND operations, their in-situ logic memory and reset latency in two transistors and one memristor integrated circuits.** different sequence of input signals and corresponding output signals demonstrating logic **a** OR or **b** AND operation and insets showing corresponding schematic diagrams of the 2T1M circuit configurations where the transistor gate voltages ($V_G$) are inputs and current is output signal ($I_{out}$); logic output signals during application of input signals and afterwards when the inputs are set to 0 V for **c** OR logic operation in the circuit diagram shown in the inset of (**a**) and **d** AND logic operation in the circuit diagram shown in the inset of (**b**). Insets on the right panels of (**c**, **d**) show the truth tables of logic output during procession of the inputs and memory afterward for all input combinations for OR and AND logic gates, respectively. Here, $V_G$ of −2 V and 3 V are considered as logic input "0" and "1", respectively. The drain voltage ($V_D$) was kept fixed at 4 V during all logic inputs as well as for logic memory. The dashed lines show the threshold current of 4 nA to distinguish between logic output "0" and "1". For the reset latency test, the decay of current is monitored during the reset process by resetting both transistors' $V_G$ to −2 V after stabilization of $I_{out}$ at each logic memory test for both logic operations.

diagnostic parameters: heart-rate and/or blood-pressure. The pre-condition is set in the phase of $V_D$, and thus either logic AND or OR applies. Real-time diagnostic parameters are mapped to logic value inputs "0" and "1" (normal and elevated ranges of blood-pressure/heart-rate, respectively). As illustrated schematically in Fig. 5c, for

healthy individuals (AND logic), both diagnostic parameters need to be "high" to trigger a medical emergency warning. In contrast, for a heart-disease patient (OR logic), a medical emergency is triggered by the emergence of at least one elevated dynamic input. In all other cases, the person is considered to be in a normal state. Using this diagnosis

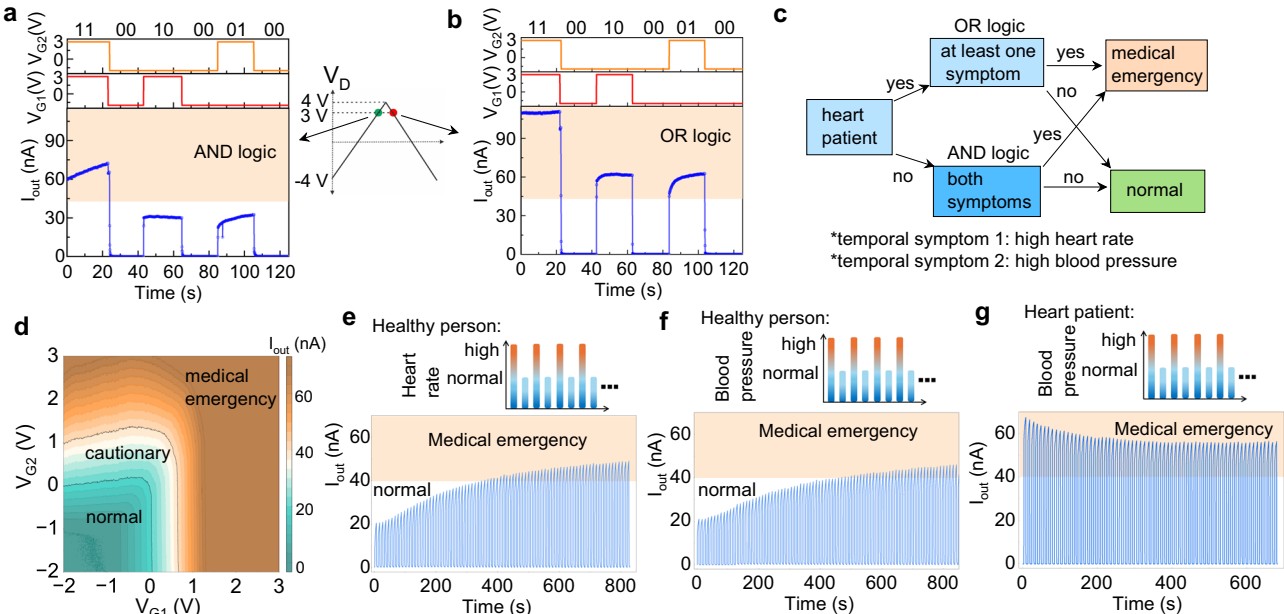

**Fig. 5 | Reconfigurability of logic operations and its illustration for a healthcare task in a single circuit of two transistors and one memristor.** reconfigurable logic operation between **a** AND and **b** OR logic at $V_D = 3$ V after a $V_D$ sweep from −4 to 3 V and 4 to 3 V, respectively, for a 2T1M circuit configuration shown in the inset of Fig. 4a; **c** a diagnosing algorithm to identify a person's health condition considering a heart patient as static input and a high heart rate and a high blood pressures as non-static or temporal inputs, **d** 2-dimensional current ($I_{out}$) map in the full range of input voltages ($V_G$) from −2 to 3 V assuming a normal-to-high range of heart rate (x-axis) and blood pressure (y-axis) at $V_D = 3$ V indicating the device can be utilized as an analog type operator with three output cases (normal, cautionary and emergency cases); transition from normal to medical emergency case for a healthy person (static input) **e** if blood pressure remains normal but heart rate continuously fluctuates and **f** if heart rate remains normal but blood pressure continuously fluctuates; **g** remaining at medical emergency condition for a heart patient (static input) if heart rate remains normal but blood pressure continuously fluctuates.

algorithm, a person's health status can be continuously monitored by comparing $I_{out}$ against the predefined threshold current.

For the realization of an analog-type operation of the circuit, $I_{out}$ is mapped continuously by sweeping $V_{G1,2}$ between the initially used logic "low" and "high" voltage values (Fig. 5d). Here, the map can be divided in three main regions, labeled "normal", "cautionary", and "medical emergency", depending on the two diagnostic parameters $V_{G1,2}$. The analog-type operation with continuous current variation enables multi-level evaluation with more intermediate health-assessments, such as "low", "medium", and "high" cautionary assessments. Combining the reconfigurable logic-type operation with the analog-type operation, dynamic health monitoring is possible. For instance, consider a scenario where either of the two diagnostic parameters fluctuates rapidly between "high" and "normal" within short time intervals. Such a scenario is depicted in Fig. 5e, f for heart-rate and blood-pressure, respectively, in which $I_{out}$ is recorded over time for alternate "high" and "normal" of the diagnostic parameter. Initially, $I_{out}$ remains low, indicating a non-critical state. However, after a series of "high" levels of the diagnostic parameter, $I_{out}$ crosses a defined threshold, triggering a medical emergency alert, an appropriate response to sustained abnormal conditions. In addition, the threshold-crossing point occurs at a different number of "high" levels between the two diagnostic parameters. This time difference reflects the varying urgency associated with these parameters, which could offer valuable insights for differential diagnosis. Next, the impact of dynamic input fluctuation is examined for a heart-patient (Fig. 5g). Here, $I_{out}$ remains above the threshold current at all "high" and "normal" levels, maintaining a continuous medical emergency condition. These results demonstrate that our devices are capable of performing complex decision-making tasks, as exemplified through a physical health diagnostic model that integrates and responds to three physiological indicators.

This highlights the potential of the system for real-time, adaptive health monitoring applications.

## Outlook

We have demonstrated polymorphic functionalities of a LAO/STO heterostructure-based single nanowire via the manipulation of the q2-DEG at room temperature. These functionalities can be integrated to form complex circuits for applications in reservoir computing, neuromorphic systems, and reconfigurable synaptic logic. The polymorphism of the device enables efficient implementations of multiple functionalities within a single device platform, which is especially relevant at the IC level with the advantages of its ease of fabrication and scalability. Our LAO/STO-based polymorphic oxide nanostructures thus pave the way for fully oxide-based monolithic ICs combining conventional sequential and non-conventional computing architectures. Its compatibility with Si-technology further supports the development of hybrid CMOS-oxide architectures that incorporate memristive and memcapacitive functionalities with standard circuit elements. In particular, memcapacitor-based artificial neural networks enable ultra-low energy consumption in computing technologies[33]. Alternatively, energy-efficient implementations such as spintronic logic devices can be realized using our LAO/STO system[61], offering significantly improved scalability and logic density over conventional CMOS technology, leveraging its strong spin-orbit coupling. Our lateral side gate approach enables tunable control of the superconducting state[62,63], offering potential applications in Josephson junctions[42]. Also, taking advantage of the lateral structure, one can add complexity in the device either electrically or optically for optoelectronic functionalities. Finally, the runtime programmable functionalities within a simple oxide-based device, combined with their versatile applications, establishes a promising pathway towards fully integrable energy-efficient polymorphic and neuromorphic computing devices.

## Methods

### Device fabrication

The devices were fabricated in three main steps. In the first step, a $TiO_2$-terminated (001)-oriented STO substrate was spin-coated with negative photoresist. Then the device layout was patterned on the surface of the substrate using electron beam lithography, followed by resist development. In the next step, 11-nm $SiO_2$ was deposited by electron beam evaporation, and then a liftoff process was carried out to create a well-defined structure. At last, pulsed laser deposition (PLD) was employed to grow 6 u.c. of LAO on the surface by ablating a single crystalline LAO target at a frequency of 1 Hz using a KrF excimer laser ($\lambda = 248$ nm) at a substrate temperature of 780 °C and an oxygen partial pressure of $1 \times 10^{-3}$ mbar followed by annealing at 500 °C for 1 h in 500 mbar of oxygen pressure. Finally, the device is ready, consisting of a nanowire and two side-gates where the q2-DEG is formed, while other regions remain insulating due to the growth of amorphous LAO on the previously deposited $SiO_2$. The details of the device fabrication process can be found in ref. 41. If not specified otherwise, the width of the investigated wire and the wire-gate distance are around 100 nm and 400 nm, respectively. The width of the gate is 1 µm.

### Device characterization

At first, ultrasonic bonding with Al-wire was performed to directly contact the q2-DEG for all the electrical measurements. For current-voltage measurements, a Keithley source meter (Model: 213 Quad Voltage Source) was used, and current was evaluated by adding a resistor of 10 k$\Omega$ or 100 k$\Omega$ in the circuits and measuring the voltage drop through the resistor with a Keithley multimeter (2000 series). A lock-in amplifier (EG&G Instruments, model: 7265) was employed to measure the real and imaginary parts of the current in the circuit by applying an ac signal using a Keithley arbitrary waveform generator (Model: 3390) in addition to the dc signal and used to evaluate the capacitance value. The data were extracted using labView program. All presented measurements were conducted at room temperature and in the dark.

To investigate the FET-operation, the lateral gates are connected to an external gate voltage ($V_G$). A bias voltage is applied to the top contact (designated as drain) while the bottom contact (source) is connected to the common ground. To switch the device functionality from transistor- to memristive operation, the laterally defined side-gates are set on a floating potential as illustrated schematically in Fig. 1a. Capacitance measurements were performed between the channel and one of the lateral gates, while the other gate was kept floating, as shown in the bottom right panel of Fig. 1a.

For physical RC application utilizing a 1T1MC integrated circuit, input voltage pulse is applied to the gates of the transistor, while the output voltage is acquired from the memcapacitor, as illustrated in Fig. 2a. To demonstrate the pattern recognition of monochrome digit images using the circuit, the gray and white pixels are assigned to input pulse amplitudes of 3 and −2 V, respectively, with a fixed pulse width of 150 ms.

The schematic of the 1T1M structure for synaptic operation is shown in Fig. 3a, where a presynaptic voltage ($V_G$) is applied to one gate of the transistor, and the PSC is measured from the memristor. The other transistor gate is left floating to facilitate charge trapping and de-trapping, thereby modulating the conductivity to emulate the operation of a synaptic transistor[64].

For the OR gate operation, two transistors are connected in parallel with one memristor in series, while for the AND gate, two transistors are connected in series with the memristor, as illustrated in the insets of Fig. 4a, b, respectively. A drain voltage ($V_D$) of 4 V was kept fixed during all the logic operations and their memory investigation. The reconfigurability of logic operations between the AND and OR gates is realized by considering another threshold current of 40 nA to distinguish between logic output "0" and "1".

## Data availability

The data that support the findings of this study are available from the corresponding authors upon request.

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

## Acknowledgments

The authors gratefully acknowledge financial support by the Deutsche Forschungsgemeinschaft (DFG, German Research Foundation) through the Würzburg-Dresden Cluster of Excellence ct.qmat - Complexity, Topology and Dynamics in Quantum Matter (EXC 2147, project-id 390858490) as well as through the Collaborative Research Center SFB 1170 "ToCoTronics" (project-id 258499086). VLR acknowledges the support from Conselho Nacional de Desenvolvimento Científico e Tecnológico (CNPq-Brasil), Proj. 311536/2022-0 and FAPESP Projs. 2025/04805-0 and 2025/00677-8.

## Author contributions

F.H. and S.H. initiated and guided the study. M.Spring, J.G., and B.L. grew the sample in discussion with M.Sing and R.C., S.K., and M.K. fabricated the devices. K.M. initiated the experiment, S.P. designed and conducted all the experimental work in discussion with F.H.. S.P, F.H., V.L., and S.H. analyzed and interpreted the experimental results. S.P. and F.H. wrote the manuscript, with input from all coauthors.

## Funding

## Competing interests

The authors declare no competing interests.
