## [Transparent Peer Review file · Nature Communications]

Oxide Interface-Based Polymorphic Electronic Devices for Neuromorphic Computing

Corresponding Author: Dr SOUMEN PRADHAN

Version 0:

Reviewer comments:

Reviewer #1

(Remarks to the Author)

In the manuscript entitled "Oxide Interface-Based Polymorphic Electronic Devices for Neuromorphic Computing", the authors study LAO/STO-based electronic devices, combining transistor, memristor, and memcapacitor functionalities into one platform for various applications, including reservoir computing, synaptic plasticity, logic computation, and reconfigurable synaptic logic. This manuscript is interesting and well-written with logical discussion and presentation. However, some detailed device parameters are missing, as well as the reliability tests (retention, endurance, variability) and long-term environmental stability. Currently, the manuscript is more toward proof-of-concept rather than a robust technological advancement. Strengthening these points can significantly enhance the quality of the manuscript and broaden the impact of this work:

1. The performance of the transistor is not carefully evaluated. What is the mobility, on/off, and subthreshold swing? These parameters are fundamental for transistor devices to evaluate its competitiveness.
2. The resistance ratio (R_{high}/R_{low}) of about ~ 5567 is good, but is not "very high". I suggest not using overstated term. Also, what are the net values of R_{high} and R_{low} ? These values are hard to see in Fig. 1c and must be given in the manuscript.
3. In the reservoir computing part, the device consists one transistor and one memcapacitor, and the authors mention that there is charge leakage from the memcapacitor which leads to short-term memory. Can long-term memory be realized in such integrated configuration? Please clarify.
4. The memcapacitive effect is attributed to charge trapping/detrapping, but no direct physical evidence is presented, which should be strengthened.
5. In Fig. 2h, the authors mention that the average VO after stimulation, along with error bars, indicates that almost all configurations can be reliably distinguished. However, it seems several output VO are at similar levels, such as (0101,1001) and (1101,1011). How to distinguish them?
6. In the synaptic plasticity part, with 1T1M configuration, the authors demonstrate the storage of information over extended periods and the transition from STP to LTP. What's the retention time for the LTP?
7. There are some typos in the manuscript, such as from -4 to 3 and 4 to 3 V. Please check carefully.
8. It is suggested to carry out endurance/variability/stability tests, which may strengthen the impact of this work.

Reviewer #2

(Remarks to the Author)

I co-reviewed this manuscript with one of the reviewers who provided the listed reports. This is part of the Nature Communications initiative to facilitate training in peer review and to provide appropriate recognition for Early Career

Researchers who co-review manuscripts.

Reviewer #3

(Remarks to the Author)

In this manuscript, the authors present a polymorphic oxide interface device based on LAO/STO that operates as a transistor, memristor, and memcapacitor, and integrate those functions for reservoir computing, synaptic plasticity, and reconfigurable logic. The topic is timely for neuromorphic and oxide-electronics communities. Several aspects would benefit from clarification, additional controls, and presentation refinements to strengthen the mechanistic interpretation and the engineering relevance.

1. Please report statistics (mean \pm SD) for threshold voltage, on/off ratio, and hysteresis area after thousands to tens of thousands of mode toggles under controlled temperature and humidity. Device-to-device and chip-to-chip distributions ($n \geq 20$) would help quantify process variability.
2. An energy analysis comparing memcapacitor-based reservoir computing with memristor- and transistor-based baselines under matched operating points would be valuable. Please provide pulse energy, energy per inference, and average power, together with quantitative descriptors of reservoir dynamics, such as time constants, memory kernels, and state-space rank, as functions of pulse width, period, and inter-pulse interval.
3. For the 2T1M logic circuits, characterization of noise margins and temperature dependence around the 4 nA decision threshold would clarify robustness. Misclassification rates under injected supply fluctuation, readout noise, and process variation, summarized via ROC curves or confusion matrices, would further support reliability.
4. Extension of in-situ logic-memory measurements beyond 300 s with retention-time distributions, reset-latency statistics, and cycling endurance (10^3 – 10^4 cycles) would define non-volatile behavior and operational limits, including any drift.
5. Controlled perturbations that vary sweep rate and direction, oxygen ambience, and optical illumination, complemented by frequency-dependent C–V analysis and activation-energy extraction, would help separate contributions from charge trapping and oxygen-vacancy migration. A concise compact model or ion-drift framework with fitting quality and residuals would increase mechanistic clarity.
6. A practical roadmap for the LAO/STO lateral side-gate architecture, such as process temperature budgets, interconnect and passivation choices, minimum line/space and channel length, projected array density, yield considerations, and CMOS back-end compatibility would clarify scalability.

Reviewer #4

(Remarks to the Author)

This manuscript presents polymorphic electronic devices based on patterned SiO₂/LaAlO₃/SrTiO₃ heterostructures in an array configuration. The topic is highly interesting, and the proposed device architecture appears both feasible and novel, making the results promising. However, the manuscript currently lacks sufficient validation regarding the key advantages and operational mechanisms of the polymorphic devices. As this aspect is critical to assessing the true impact of the work, I unfortunately cannot recommend the manuscript for publication in Nature Communications.

1. The polymorphism of electronic devices is highly intriguing, yet further validation is required to determine whether it truly represents a technological advancement in terms of efficiency. Although enabling a single device to perform multiple roles can potentially improve integration density, it is also worth noting that in conventional ICs, individual components such as transistors, memristors, and capacitors typically operate concurrently rather than sequentially. Therefore, the benefit gained from consolidating multiple functionalities into a single device may not be as substantial as suggested. An objective discussion on this point would strengthen the manuscript. I recommend citing recent research progress in this field with appropriate references. Additionally, and more importantly, the manuscript should include a quantitative analysis that clearly states the energy efficiency advantage achieved by employing polymorphism in a single device, compared to using multiple devices, specifically for the proposed device in this study.
2. In Fig. 1a, the STO substrate is shown underneath the LAO layer for the LAO/STO structure, while in the LAO/SiO₂ notation, LAO appears to be located below SiO₂ instead. Please confirm whether this interpretation is correct, and revise the notation to ensure consistency throughout the manuscript. Most importantly, a concise description of the materials, device structure, and fabrication process used in this work should be included in the main text, not only in the Methods section. Based on the current figures and descriptions, it is not clearly understood where the active channel is located and which region functions as the gate dielectric layer. This should be clarified in the manuscript.
3. The manuscript attributes most of the observed resistance changes to oxygen vacancy migration, yet provides very limited explanation or validation of the actual device mechanism. In reality, even in the case of ideal single-crystalline LAO film, atomic interdiffusion or chemical reactions may cause resistance variation, which could lead to irreversible damage during long-term device operation. Furthermore, when SrTiO₃ (STO) is used as the substrate, La diffusion into the substrate can also result in the formation of a metallic La-doped STO interface. Such changes may degrade the gating efficiency over time. Additionally, even if we assume, based on prior studies, that the observed resistance changes can be attributed solely to oxygen vacancy redistribution, the manuscript lacks sufficient validation to support the claim that variations in capacitance or

hysteresis characteristics are also caused simply by oxygen vacancies. Although demonstrating the applicability of this heterostructure to reservoir computing is indeed the key highlight of this paper, the current level of understanding of the material system appears insufficient to justify the claimed application potential. A deeper investigation and discussion regarding the materials and mechanisms are necessary.

4. In hardware reservoir computing (RC), the validity of the implementation critically depends on how effectively the device exploits both “nonlinearity” and “short-term memory (fading memory)” as computational resources. These properties enable input mapping into a higher-dimensional state space with fading temporal dynamics. In Fig. 2, although the device indeed exhibits certain nonlinear and memory behaviors, the current results make it difficult to conclude that a genuine reservoir has been realized. The observed performance appears consistent with simple charge accumulation and retention rather than a true high-dimensional state transformation. Specifically, (1) the reservoir size remains too limited to demonstrate meaningful state expansion; (2) the classification task seems to rely mainly on distinguishable output voltage (V_o) levels rather than trained readout weights applied to high-dimensional reservoir states; (3) Temporal memory is not quantified to evaluate memory capacity; and (4) a comparison between conditions with and without nonlinearity is missing, making it unclear whether nonlinearity genuinely contributes to computation. The authors should clarify the role of nonlinear dynamics and fading memory more explicitly and provide stronger evidence that the demonstrated function surpasses simple state-dependent thresholding.

5. While multiple functionalities (RC, synaptic plasticity, and logic operations) are demonstrated in Figs. 3,4, and 5, they all rely on the same underlying charge-trapping mechanism. As a result, the current demonstrations appear more like concept-level application repackaging rather than revealing distinct and functionally independent polymorphic behaviors. To strengthen the scientific contribution, the manuscript should examine either:

- (i) how these functionalities do not interfere with each other (e.g., cycling one mode does not degrade another), or
- (ii) clear quantitative benefits over using separate devices (energy, area, retention, tunability, etc.).

Without such deeper validation, it is difficult to evaluate the practical or scientific significance of the claimed polymorphism.

Version 1:

Reviewer comments:

Reviewer #1

(Remarks to the Author)

The authors have carefully addressed my previous concerns and provided additional results, including transistor performance parameters, endurance, retention, and long-term stability tests. Also, the mechanism explanations for the memristive and memcapacitive behaviors are strengthened in the revised manuscript. I have no further major comments and would suggest publication of the manuscript.

Reviewer #2

(Remarks to the Author)

Reviewer #3

(Remarks to the Author)

The authors have adequately addressed all of the concerns raised during the review process. The revised manuscript is clear and technically sound, and the changes have strengthened the overall quality of the work. On this basis, I support publication of the manuscript in its present form.

Reviewer #4

(Remarks to the Author)

This manuscript reports a broad range of results, spanning from materials-science-based mechanisms for modulating resistance and capacitance in LAO/STO heterostructures with a tunable 2DEG, to device-level multifunctionality and practical applications such as reservoir computing. The authors have carefully addressed the shortcomings raised in the previous round of comments, and the advantages enabled by polymorphism are now articulated more effectively. Overall, the manuscript has reached a level that is suitable for publication. Nevertheless, addressing the following points would further strengthen the quality and clarity of the work.

(1) Regarding efficiency, the intended meaning does not appear to be that each individual device operates at a lower power while achieving comparable functionality. Rather, the efficiency advantage seems to arise at the integrated-circuit (IC) level, where polymorphism enables multiple functionalities within a single device platform, leading to higher system-level efficiency compared to integrating multiple distinct types of devices. This distinction should be stated more explicitly and consistently throughout the manuscript.

(2) The I–V and C–V measurements (Figs. R19 and R20) are generally very clean and well measured; however, the near absence of noise raises questions about the level of device-to-device and measurement-to-measurement reproducibility.

While some degree of data selection is understandable for clearly conveying the main message, adding a discussion on reproducibility across different devices and repeated measurements would make the study more objective and convincing. If possible, the authors may also consider including error bars, such as standard deviations, in the relevant figures to quantitatively illustrate the reproducibility.

Version 2:

Reviewer comments:

Reviewer #4

(Remarks to the Author)

The authors have addressed all of my comments adequately. Now the manuscript is ready to be published in Nature communications.

Reviewer #1:

Reviewer #1 (REMARKS to AUTHOR(s)):

In the manuscript entitled “Oxide Interface-Based Polymorphic Electronic Devices for Neuromorphic Computing”, the authors study LAO/STO-based electronic devices, combining transistor, memristor, and memcapacitor functionalities into one platform for various applications, including reservoir computing, synaptic plasticity, logic computation, and reconfigurable synaptic logic. This manuscript is interesting and well-written with logical discussion and presentation. However, some detailed device parameters are missing, as well as the reliability tests (retention, endurance, variability) and long-term environmental stability. Currently, the manuscript is more toward proof-of-concept rather than a robust technological advancement. Strengthening these points can significantly enhance the quality of the manuscript and broaden the impact of this work.

Authors' reply: We sincerely thank the Reviewer for the positive evaluation of our work and constructive feedback on it. We appreciate the recognition of the manuscript's logical structure and scientific interest. Below, we provide detailed point-by-point responses to each of the Reviewer's comments.

Comments:

1. The performance of the transistor is not carefully evaluated. What is the mobility, on/off, and subthreshold swing? These parameters are fundamental for transistor devices to evaluate its competitiveness.

Authors' reply:

We thank the Reviewer for this insightful comment. Herewith we clarify the transistor performance parameters of our devices.

The mobility of the quasi-two-dimensional electron gas (q2-DEG) formed at the LaAlO₃/SrTiO₃ (LAO/STO) interface in our devices is approximately 5.44 cm²/(Vs) at room temperature and the temperature dependence of mobility and carrier density (from 2 K to 300 K) has been systematically reported by our group in Ref. A, showing the expected increase in mobility and decrease in carrier density with reducing the temperature. These values are consistent with the literature (Ohtomo et al., Ref. [B]) for high-quality LAO/STO interfaces [C]. The value is comparable to, or even exceeds, those reported for other oxide-based 2DEG systems such as LaTiO₃/STO [D], γ -Al₂O₃/SrTiO₃ [E], and GdTlO₃/STO [F]. Moreover, in combination with its tunability (via gating), rich emergent phases (superconductivity, magnetism, Rashba spin-orbit coupling), and mature growth and processing techniques, LAO/STO highlights the potential of oxide electronics for prototyping multifunctional oxide devices.

Next, to evaluate the performance of LAO/STO-based device in transistor configuration, we measured the drain current (I_D) as a function of gate voltage (V_G) over a range of -2 and 3 V for different drain biases (V_D). Figure R1(a) displays the corresponding transfer characteristics, showing an on/off current ratio (I_{on}/I_{off}) in the range of $\sim 10^5$ confirming robust gate modulation. The subthreshold swing (SS) values extracted using the relation [$SS = \partial V_G / \partial \log(I_D)$] are presented in Fig. R1(b). Remarkably, our devices exhibit SS values below the theoretical limit of 60 mV/decade at room temperature, i.e. a sub-thermal switching mechanism.

To elucidate the physical origin of this behavior, we developed a theoretical model (Ref. [G]) that provides a generalized analytical expression for SS in memory-assisted transistors. The model captures the key experimental features and demonstrates how intrinsic memory effects in oxide heterostructures enable sub-thermal switching by passing the conventional Boltzmann limit. We are currently conducting an in-depth experimental study on this sub-thermal behavior which will be the focus of a dedicated publication. For this reason, we refrain from disclosing explicit SS values below the 60 mV/decade limit in the current manuscript, as this topic constitutes an independent, ongoing investigation.

Fig. R1: (a) Transfer characteristic curves for different V_D values for the LAO/STO device in transistor configuration. (b) Calculated subthreshold swing (SS) values with I_D for the same V_D values using the formula [$SS = \partial V_G / \partial \log(I_D)$] highlighting the theoretical limit of 60 mV/decade at room temperature.

References:

- [A] Maier, Patrick, et al. "Gate-tunable, normally-on to normally-off memristance transition in patterned LaAlO₃/SrTiO₃ interfaces." *Applied Physics Letters* 110.9 (2017).
- [B] Ohtomo, A., and H. Y. Hwang. "A high-mobility electron gas at the LaAlO₃/SrTiO₃ heterointerface." *Nature* 427.6973 (2004): 423-426.
- [C] Förg, B., Christoph Richter, and Jochen Mannhart. "Field-effect devices utilizing LaAlO₃-SrTiO₃ interfaces." *Applied Physics Letters* 100.5 (2012).
- [D] Leng, Huaqian, et al. "Competing conduction mechanisms for two-dimensional electron gas at LaAlO₃/SrTiO₃ heterointerfaces." *Applied Physics Letters* 124.18 (2024).
- [E] Niu, Wei, et al. "Suppressed carrier density for the patterned high mobility two-dimensional electron gas at γ -Al₂O₃/SrTiO₃ heterointerfaces." *Applied Physics Letters* 111.2 (2017).
- [F] Son, Junwoo, et al. "Epitaxial SrTiO₃ films with electron mobilities exceeding 30,000 cm² V⁻¹s⁻¹." *Nature materials* 9.6 (2010): 482-484.
- [G] Silva, Rafael Schio Wengenroth, et al. "2D Canonical Approach for Beating the Boltzmann Tyranny Using Memory." *arXiv preprint arXiv:2510.24883* (2025).

Heeding the Reviewer's suggestion, the following sentences have been added to the revised version of the manuscript: "The mobility of the q2-DEG formed at the LAO/STO interface in our devices is approximately $5.44 \text{ cm}^2/(\text{Vs})$ at room temperature. However, its temperature variability was reported in our old publication" (line# 114-117).

"The results demonstrate an on-current in the μA range with on/off current ratio in the range of $\sim 10^5$ confirming robust gate modulation" (line# 126-127).

2. The resistance ratio ($R_{\text{high}}/R_{\text{low}}$) of about ~ 5567 is good, but is not "very high". I suggest not using overstated term. Also, what are the net values of R_{high} and R_{low} ? These values are hard to see in Fig. 1c and must be given in the manuscript.

Authors' reply:

We thank the Reviewer for this valuable observation and for pointing out the need for clarification. The resistance values are $R_{\text{high}} = 8.52 \text{ G}\Omega$ and $R_{\text{low}} = 1.53 \text{ M}\Omega$. The term "very high" has been replaced with a more accurate description in the revised manuscript. The net resistance values have also been explicitly added to the main text for clarity.

Heeding the Reviewer's suggestion, the following sentence has been added to the revised version of the manuscript: "However, two resistance values of $8.52 \text{ G}\Omega$ (R_{high}) and $1.53 \text{ M}\Omega$ (R_{low}) extracted from the linear fits of the experimental forward and reverse bias data near zero bias yields a resistance ratio of approximately 5.6×10^3 for the device." (line# 161-163).

3. In the reservoir computing part, the device consists of one transistor and one memcapacitor, and the authors mention that there is charge leakage from the memcapacitor which leads to short-term memory. Can long-term memory be realized in such integrated configuration? Please clarify.

Authors' reply:

We thank the Reviewer for this important question and the opportunity to clarify the memory characteristics of the 1T1MC configuration.

In our device architecture, the crystalline LAO/STO interface serves as the conducting channel, while the two lateral gates are separated by a SiO_2 dielectric layer, as shown schematically in Fig. R2. Due to the relatively low dielectric constant of SiO_2 (~ 3.9), the electric field lines spread laterally, weakening the vertical field confinement. This promotes field-induced charge leakage from the drain region, which manifests as short-term memory in the present configuration. To enhance retention and enable long-term memory, two key design modifications can be considered in the future:

- (i) increasing the lateral separation between the drain and gate contacts, and
- (ii) replacing SiO_2 with a high-k dielectric such as HfO_2 .

High-k materials provide stronger vertical electrostatic coupling and improved gate control, thereby effectively suppressing leakage and stabilizing charge trapping. These changes are expected to extend the memory retention time significantly in similar integrated architectures.

Fig.R2: Cross-sectional schematic diagram of the device, in which crystalline LAO (c-LAO) growth directly on STO results in formation of quasi-2-dimensional electron gas (q2-DEG) at the interface while LAO grown on SiO₂ regions resulting in amorphous growth (a-LAO).

Heeding the Reviewer's query, the following sentence has been added to the revised version of the manuscript: "This leakage can be controlled by increasing the separation between the gate and channel and also by replacing SiO₂ with a high- κ dielectric material as gate oxide layer." (line# 228-230).

4. The memcapacitive effect is attributed to charge trapping/detrapping, but no direct physical evidence is presented, which should be strengthened.

Authors' reply:

We thank the Reviewer for this valuable comment and the opportunity to clarify the physical origin of the memcapacitive behavior. As described in the manuscript, the capacitance measurements were carried out between the drain contact and one lateral gate, while the other lateral gate was left floating. To further substantiate the mechanism, we performed additional control experiments by sweeping the drain voltage (V_D) with the auxiliary gate grounded, keeping all other parameters identical.

The corresponding C-V curves, measured at an AC excitation frequency of 10 Hz, are shown in Fig. R3(a). When the lateral gate is grounded, any charge reaching the gate is immediately drained, resulting in a non-hysteretic C-V response. In contrast, under the floating gate configuration, localized charge trapping and detrapping at the lateral gate leads to a pronounced hysteresis in the full-cycle C-V characteristic confirming that charge localization dynamics are responsible for the observed memcapacitive behavior.

This experimental observation is further supported by a theoretical model that reproduces the memcapacitance response, as shown in Fig. R3(b). The model includes contributions from the geometric oxide capacitance (incorporating also interface states), a bias-dependent depletion capacitance, and a dynamic term accounting for the charging and discharging of the floating gate. The combined effect of these elements explains the hysteretic capacitance response and its dependence on frequency and dielectric properties. A detailed description of this dynamic model and its correlation with experimental conditions can be found in Ref. A below.

Fig. R3: (a) Full cycle C-V curves for V_D sweep between ± 1 V with the auxiliary gate grounded and at floating condition measured at an AC voltage of 10 Hz. (b) Theoretically simulated capacitance loop of the device using a dynamic model for trapping charges at the floating gate contrasting two values of the oxide layer capacitance.

References:

[A] Pradhan, S., Lopez-Richard, V., Hartmann, F., Silva, A.L.C., Castelano, L.K., Spring, M., Kuhn, S., Sing, M., Claessen, R., Höfling, S., et al.: Gate controlled analog memcapacitance in $\text{LaAlO}_3/\text{SrTiO}_3$ interface-based devices, arXiv preprint arXiv:2512.11176 (2025)

Heeding the Reviewer's suggestion, the following sentence has been added to the revised version of the manuscript: "A comprehensive theoretical framework capturing the capacitance modulation through charge localization dynamics on the floating gate, consistent with the experimental C-V characteristics, is presented in more detail in (Pradhan, Soumen et al., Gate-controlled analog memcapacitance in $\text{LaAlO}_3/\text{SrTiO}_3$ interface-based devices, arXiv preprint arXiv:2512.11176 (2025))." (line# 179-182).

5. In Fig. 2h, the authors mention that the average VO after stimulation, along with error bars, indicates that almost all configurations can be reliably distinguished. However, it seems several output VO are at similar levels, such as (0101,1001) and (1101,1011). How to distinguish them?

Authors' reply:

We thank the Reviewer for this insightful comment. Indeed, as correctly noted, a few configurations such as (0101, 1001) and (1101, 1011) exhibit overlapping output voltages (V_o) within the error-bar range when all 16 types of 4-bit pulse trains are analyzed. However, for the digit classification task presented in the manuscript, only six 4-bit combinations: (1111), (1001), (0010), (0001), (1000), and (1010) are relevant. These configurations, which correspond to the monochrome representations of digits 0-9, are clearly distinguishable in our experimental data. Therefore, the classification of the ten digits using our 1T1MC device architecture can be performed without ambiguity or error.

Fig. R4: (a) Simulated input pulse at V_g and the control parameters tested for the implemented optimization models. (b) Pulse sequence corresponding to the numbers (0101) and (1001). (c) Output for two contrasting constant pulse widths. (d) Output for a constant pulse width (red) and one (blue) generated by a logarithmic dependence on the digit position. (e) Contrast ratios for the three pulse configurations displayed in panels (d) and (c).

We agree that when considering all 16 possible 4-bit pulse trains, partial overlap between some output states limits the general pattern classification capability. To address this, we are currently developing an optimization model that aims to enhance output resolution by tuning pulsing parameters, as illustrated in Fig. R4(a), such as pulse width t_j , inter-pulse gap t_j^0 , and amplitude V_j . For clarity, we present here an example comparing two specific input sequences, (0101) and (1001), highlighted by the Reviewer. Figure R4(b) shows the corresponding input pulse trains, and Fig. R4(c) displays the resulting outputs for two distinct constant pulse widths: $t_j = \tau$ and $t_j = \tau/2$. As demonstrated in Fig. R4(d), output distinguishability can be significantly improved by adopting a variable pulse width generated through a logarithmic dependence on the digit position, comparing: $t_j = \tau$ with $t_j = \tau \ln(kj)$. To quantify how well the outputs corresponding to different input digits of size j , $V_0(j, \text{digit})$, can be distinguished, we define the contrast ratio

$$\varepsilon[t_j] = 2[V_0(j, \text{digit}2) - V_0(j, \text{digit}1)] / [V_0(j, \text{digit}2) + V_0(j, \text{digit}1)].$$

The evolution of this contrast ratio as a function of the pulse width is shown in Fig. R4(e). This analysis was performed using our theoretical emulation framework, which accurately reproduces the transistor, memristor, and memcapacitor functionalities (see, for instance, Refs. [A,B,C]). These models are currently being extended to develop optimization protocols that include pulse modulation strategies, noise sensitivity, and single-bit error resilience. The results of this ongoing study will be reported in a forthcoming publication.

References:

[A] Silva, Rafael Schio Wengenroth, et al. "2D Canonical Approach for Beating the Boltzmann Tyranny Using Memory." *arXiv preprint arXiv:2510.24883* (2025).

[B] Lopez-Richard, V. et al; "Beyond equivalent circuit representations in nonlinear systems with inherent memory". *J. Appl. Phys.* 28 (2024); 136 (16): 165103

[C] Pradhan, S., Lopez-Richard, V., Hartmann, F., Silva, A.L.C., Castelano, L.K., Spring, M., Kuhn, S., Sing, M., Claessen, R., Höfling, S., et al.: Gate controlled analog memcapacitance in LaAlO₃/SrTiO₃ interface-based devices, *arXiv preprint arXiv:2512.11176* (2025)

Heeding the Reviewer's concern, the following sentence has been added to the revised version of the manuscript: "The contrast among the outputs can be improved by tuning the parameters such as pulse width, inter-pulse gap and pulse amplitude." (line# 274-275)

6. In the synaptic plasticity part, with 1T1M configuration, the authors demonstrate the storage of information over extended periods and the transition from STP to LTP. What's the retention time for the LTP?

Authors' reply:

We thank the reviewer for this important question.

As shown in Fig. 3(c) of the main manuscript, the 1T1M configuration exhibits a clear transition from short-term potentiation (STP) to long-term potentiation (LTP) when applying repeated V_G pulses between 3 V and 0 V at higher drain voltages V_D . However, in the original version of the manuscript we did not quantify the retention time associated with the LTP state. To address this, we performed additional measurements of the postsynaptic current (PSC) under prolonged excitation and extended monitoring.

Specifically, we applied a 100 s gate pulse of $V_G=3V$ at $V_D=4V$, during which the PSC increases monotonically, consistent with synaptic potentiation (Fig. R5). Following the pulse, V_G was set to 0 V and the PSC decay was recorded for more than 2 hours. The initial relaxation shows a rapid exponential decay, characteristic of STP, and is well fitted by a double-exponential function with relaxation times of approximately 2.60 min and 12.97 min. Beyond ~50 min, however, the decay rate becomes very small, and the PSC stabilizes into a persistent long-term memory plateau that remains essentially unchanged for the full 2-hour duration of the measurement.

These results demonstrate that the 1T1M architecture supports LTP retention for at least several hours under ambient conditions.

Fig. R5: Post-synaptic current (PSC) measurement (bottom panel) for 1T1M device configuration for a longer V_G pulse of 100 s and memory of current after turning V_G to 0 as shown in top panel. The inset shows the initial exponential decay of current memory with two exponential fits.

7. There are some typos in the manuscript, such as from -4 to 3 and 4 to 3 V. Please check carefully.

Authors' reply:

We thank the Reviewer for catching these typographical errors. We have carefully proofread the entire manuscript and corrected all instances of the voltage-sweep notation. All such corrections are indicated in the tracked-changes version of the revised manuscript.

8. It is suggested to carry out endurance/variability/stability tests, which may strengthen the impact of this work.

Authors' reply:

We thank the Reviewer for the constructive suggestion to assess endurance, variability, and stability, as these aspects are indeed essential for evaluating technological relevance. In response, we have carried out additional endurance measurements on both the memristive and memcapacitive operating modes.

Endurance/variability test:

A systematic endurance test in the memristive configuration was performed by sweeping the drain voltage (V_D) between ± 5 V for 2×10^3 consecutive cycles while keeping both lateral gates floating as shown in Fig. R6(a). During the initial tens of cycles, we observe small shifts in the maximum and minimum current levels ascribed to the transient stabilization of charge

distribution at the floating gates. Importantly, after this initial settling, the hysteresis loops remain highly reproducible with only minor cycle-to-cycle variations. From the linear fit near zero, we extract R_{on} and R_{off} for each cycle. R_{off} remains stable (10-15 G Ω), while R_{on} decreases slightly within the M Ω range, leading to the gradual monotonic increase in R_{off}/R_{on} ratio over cycles as displayed in Fig. R6(b). This monotonic trend is consistent with slow charge accumulation at the floating gates when no explicit reset is applied, a mechanism completely aligned with our memristive model. Analogous trends are observed in Fig. R6(c) where the loop areas for both positive (A^+) and negative (A^-) V_D sweeps evolve in a similarly stable and monotonic manner. These results demonstrate that the device remains structurally and functionally robust over thousands of switching cycles.

Fig. R6: (a) I_D - V_D hysteresis loops in memristive configuration of the device for sweep cycles of 2×10^3 . (b) Variation of R_{off}/R_{on} value with number of sweep cycles where R_{off} and R_{on} values were calculated from the linear fit of current variation near zero bias for forward and reverse sweeps, respectively. (c) Variation of hysteresis area with cycle number for $V_D > 0$ (A^+) and $V_D < 0$ (A^-).

As the memristive and memcapacitive behaviors originate from the same underlying mechanism: controlled charge localization and release at the lateral floating gate we expect analogous stability in memcapacitive operation. To verify this, we performed 100 full C-V sweep cycles using AC excitation (20 mV, 10 Hz) while sweeping V_D . The C-V curves in Fig. R7(a) show complete overlap with no measurable drift in the hysteresis shape or magnitude. We also extracted the capacitance values at zero bias from the forward and reverse sweep

directions (C_{low} and C_{high}). Both remain constant throughout all cycles (Fig. R7(b)), confirming excellent cycle stability of the memcapacitive operation.

Fig. R7: (a) C-V hysteresis curves with V_D sweep between ± 4 V for 100 cycles keeping the auxiliary gate at floating condition measured at an AC voltage of 10 Hz. (b) Variation of C_{low} and C_{high} with cycle number extracted from forward and reverse sweeps at zero bias.

Stability test:

In our devices, the q2-DEG, formed at the crystalline LAO/STO interface is the foundation of all polymorphic functionalities. Consequently, long-term stability is determined primarily by the robustness of this interfacial electron system. As reported by Trier *et al.*, a 6-unit cell crystalline LAO exhibits negligible time-dependent drift in sheet resistance, in contrast to thinner crystalline films (≤ 4 u.c.) or amorphous LAO grown on STO [A]. Following these findings, we consistently use 6 u.c. crystalline LAO films on STO to ensure stable and reproducible q2-DEG formation. These devices have been operated for several years. As an example, Fig. R8 shows the I_D - V_D hysteresis curves measured in the memristive configuration immediately after fabrication, and again after 3 and 5 years. Although the absolute current levels decrease modestly over time, the memristive hysteresis and distinct resistive states remain clearly preserved even after 5 years, demonstrating robust long-term functional stability. Importantly, these devices were stored in open air, rather than in a high vacuum or controlled atmosphere, which further highlights their resilience.

Fig. R8: Current-voltage (I_D - V_D) hysteresis curves measured in memristive configuration of the device after fabrication, 3 and 5 years after the fabrication.

References:

[A] Trier, Felix, et al. "Degradation of the interfacial conductivity in LaAlO₃/SrTiO₃ heterostructures during storage at controlled environments." *Solid State Ionics* 230 (2013): 12-15.

The endurance test results in memristive and memcapacitive configurations are included in Figs.S5 and S6, respectively in the revised supplementary material and corresponding text is included in Supplementary Note 2. The long-term stability test is included in Supplementary Note 3 and Fig.S7.

Reviewer #2:

Reviewer #2 (REMARKS to AUTHOR(s)):

Authors' reply:

We are thankful to the reviewer and co-reviewer for reviewing our manuscript to improve it.

Reviewer #3 (Remarks to the Author):

In this manuscript, the authors present a polymorphic oxide interface device based on LAO/STO that operates as a transistor, memristor, and memcapacitor, and integrate those functions for reservoir computing, synaptic plasticity, and reconfigurable logic. The topic is timely for neuromorphic and oxide-electronics communities. Several aspects would benefit

from clarification, additional controls, and presentation refinements to strengthen the mechanistic interpretation and the engineering relevance.

Authors' reply:

We sincerely thank the Reviewer for the positive assessment of our work and for recognizing its relevance to the neuromorphic and oxide-electronics communities. We greatly appreciate the constructive comments and suggestions aimed at improving the clarity, mechanistic interpretation, and engineering significance of our study. We have carefully addressed each point through additional analyses, clarifications, and revisions, as detailed below.

Comments:

1. Please report statistics (mean \pm SD) for threshold voltage, on/off ratio, and hysteresis area after thousands to tens of thousands of mode toggles under controlled temperature and humidity. Device-to-device and chip-to-chip distributions ($n \geq 20$) would help quantify process variability.

Authors' reply:

We thank the reviewer for this valuable suggestion. In response, we conducted additional reliability and variability measurements under controlled temperature (298 K) and relative humidity (40%). These tests include cycle-to-cycle stability, threshold-voltage statistics, on/off ratio distributions, and hysteresis-area variability.

Transistor configuration:

We applied V_G sweeps between -2 V and 3 V for 1×10^3 cycles while keeping the drain voltage fixed at $V_D = 0.2$ V. The transfer characteristics remain stable over all cycles, as shown in Fig. R9. Only a small current variation and a slight leftward shift in threshold voltage (V_T) are observed for the first tens of cycles; afterwards V_T converges and remains stable. The extracted contribution yields $V_T = -0.160 \pm 0.015$ V, demonstrating excellent stability and narrow statistical spread over repeated programming cycles.

Fig. R9: I_D - V_G curves in transistor configuration over 1×10^3 cycles at $V_D = 0.2$ V.

Memristive configuration:

We next applied V_D sweeps between ± 5 V for 2×10^3 cycles with both lateral gates floating (Fig. R10(a)). The memristive hysteresis loops remain robust during cycling. Small variations in extrema occur only in the initial cycles; afterward, the device stabilizes without degradation or collapse of the hysteresis area. Notably, the current amplitude increases progressively with cycling, reflecting stable endurance rather than performance deterioration.

We extracted the low-resistance and high-resistance states (R_{on} and R_{off}) from linear fits near zero bias for backward and forward sweeps. R_{off} remains stable in the 10-15 G Ω range, while R_{on} decreases gradually in the M Ω range. The resulting R_{off}/R_{on} ratio, displayed in Fig. R10(b), shows a small initial transient followed by an almost perfectly linear, slow increase: $\sim 0.27/\text{cycle}$ increment rate with standard deviation of 103 from the regular residuals. This monotonic trend is consistent with slow charge accumulation at the floating gates when no explicit reset is applied, a mechanism completely aligned with our memristive model. We also quantified the hysteresis areas for $V_D > 0$ (A^+) and $V_D < 0$ (A^-). Both quantities exhibit similar behavior: an initial transient followed by a slow, nearly linear increase (Fig. R10(c)). From the linear fits, the increment rates of ~ 0.05 and ~ 0.04 nW/cycle with standard deviations of 4.2 and 2.83 nW are observed for A^+ and A^- , respectively. These results confirm that the memristor functionality does not deteriorate up to 2×10^3 cycles.

Fig. R10:(a) I_D - V_D hysteresis loops in memristive configuration of the device for sweep cycles of 2×10^3 . (b) Variation of R_{off}/R_{on} value with sweep cycle number where R_{off} and R_{on} values were calculated from the linear fit of current variation near zero bias for forward and reverse sweeps, respectively. (c) Variation of hysteresis area with cycle number for $V_D > 0$ (A^+) and $V_D < 0$ (A^-). The red lines in (b) and (c) represent the linear fit of the constant increment of data.

Overall these results demonstrate that the device exhibits excellent endurance, narrow statistical variability, and stable switching characteristics under repeated operation. We agree with the reviewer that device-to-device and chip-to-chip statistics ($n > 20$) would further strengthen process-variability analysis. Such studies require arrays of patterned devices,

which fall beyond the scope of the present work. This manuscript aims primarily to establish polymorphic functionality and demonstrate application-level potential. We are currently fabricating large-area chip arrays, and comprehensive statistical analyses will be reported in future publications..

The new statistics on threshold voltage, on/off ratio and hysteresis area variability are now summarized in Supplementary Note 2, with corresponding data in Figs. S4 and S5 of the revised Supplementary Material.

2. An energy analysis comparing memcapacitor-based reservoir computing with memristor- and transistor-based baselines under matched operating points would be valuable. Please provide pulse energy, energy per inference, and average power, together with quantitative descriptors of reservoir dynamics, such as time constants, memory kernels, and state-space rank, as functions of pulse width, period, and inter-pulse interval.

Authors' reply:

We thank the reviewer for this important and thoughtful suggestion. We agree that an energy comparison between the 1T1MC and 1T1M architectures, under matched operating conditions, is essential for assessing their relative efficiency for reservoir computing (RC). Below we provide a clarified and quantitative analysis, including pulse-energy estimates, inference-level energy, average power, and the main dynamical descriptors of the reservoir.

Comparison of energy consumption between 1T1MC and 1T1M device configurations:

According to the main manuscript, the series connection of one transistor with one memcapacitor or one memristor in addition to input and output signals are schematically shown in Fig. R11. In the case of one transistor, one memcapacitor (1T1MC)-based integrated circuit, we have used fixed drain voltage (V_D) and gate voltage (V_G) as input voltage pulse.

Fig. R11: Schematic representation of (a) 1T1MC and (b) 1T1M integrated circuits.

(a) 1T–1MC reservoir (memcapacitor-based)

During reservoir computing operation, a voltage $V_{MC} = 2.14$ V develops across the memcapacitor under a gate pulse $V_G = 3$ V and $V_D = 4$ V. At positive V_G , the measured memcapacitance at the memcapacitor node ~ 155 pF. Therefore, the charge stored in the memcapacitor, $Q = C_{MC} \times V_{MC}$. and the energy drawn from the fixed drain supply can be calculated as:

$$E_{Source} = \int V_D i(t) dt = \int V_D dQ = V_D \cdot Q = V_D \cdot C_{MC} \cdot V_{MC} = 4 \times 155 \times 10^{-12} \times 2.14 \text{ J} = 1.32 \text{ nJ}$$

On the other hand, energy drawn from the transistor gate driver (considering both the charging and discharging pulse) must include the standard capacitive charging+discharging term:

$$E_{Gate} = C_G \cdot V_G^2 = 155 \times 10^{-12} \times 3^2 \text{ J} = 1.39 \text{ nJ}$$

Thus, the total energy drawn per pulse for 1T1MC configuration is

$$E_{pulse, total} = E_{Source} + E_{Gate} = (1.32 + 1.39) \text{ nJ} = 2.71 \text{ nJ}$$

(b) 1T–1M reservoir (memristor-based)

Now, let us evaluate the energy consumption for one transistor, one memristor (1T1M)-based reservoir computing system. For comparison, we consider the same supply voltage for a single input pulse. As shown in Fig. 3(b) in the main manuscript, the output current (I_D) reaches $\sim 2.53 \times 10^{-8}$ A for a single 150 ms V_G pulse at $V_D = 4$ V

Here, the energy drawn from the V_D supply is:

$$E_{Source} = \int_0^\tau V_D i(t) dt = V_D \cdot I_D \cdot \tau = 4 \times 2.53 \times 10^{-8} \times 0.15 \text{ J} = 15.18 \text{ nJ}$$

The gate driven energy remains:

$$E_{Gate} = 1.39 \text{ nJ}$$

And the total energy drawn per pulse is

$$E_{pulse, total} = E_{Source} + E_{Gate} = (15.18 + 1.39) \text{ nJ} = 16.57 \text{ nJ}$$

Therefore, the memcapacitor-based reservoir consumes approximately six times less energy per pulse than the memristor-based reservoir under identical bias conditions.

Energy and power analysis in 1T1MC-based RC system:

Using the 4-bit input scheme (four pulses per inference):

$$E_{inference} = 2.71 \times 4 \text{ nJ} = 10.84 \text{ nJ}.$$

The inference rate is

$$r = \frac{1}{4T} = 1.667 \text{ s}^{-1}$$

Average power: $P_{av} = E_{inference} \times r = 18.07 \text{ nW}$

Fig. R12: (a) Output voltage (V_O) with its memory for a single input pulse of varying width at fixed V_D of 4 V for 1T1MC device, (b) a short window of V_O decay as indicated by the dotted rectangle in Fig. (a) with corresponding fits using two exponential decay functions, (c) the variation of two decay constants with pulse duration.

Time constant and memory kernel analysis:

The reviewer requested dynamical metrics of the reservoir dynamics. In response, we extracted the characteristic memory kernels, decay time constants directly from our time-resolved output-voltage (V_O) measurements under different pulse-width stimuli. As shown in Fig. 2(c) in the main manuscript, V_O measured after a single input pulse of varying width exhibits (i) an abrupt drop immediately after pulse termination followed by (ii) a multi-timescale relaxation tail that constitutes the device's short-term memory (Fig. R12a). A zoomed view of the relaxation (Fig. R12b) clearly reveals two distinct decay regimes. The memory kernel increases with increasing the pulse width. To quantitatively extract the memory kernels, we fitted the relaxation to a double-exponential function

$$V_O = A_1 + A_2 \exp(-t/\tau_1) + A_3 \exp(-t/\tau_2)$$

where τ_1 and τ_2 represent the fast and slow decay constants, respectively. The fits are illustrated in Fig. R13(b). The extracted time constants versus pulse width are plotted in Fig. R12(c). The fast component τ_1 increases monotonically with pulse width, meaning that the short-term memory kernel becomes progressively longer. This is a desirable feature in physical reservoirs, where input amplitude-dependent fading memory broadens computational richness. The slow component, τ_2 , remains nearly constant for short pulses but exhibits a sharp increase for pulse width approaching 0.9 s, indicating the onset of a second, slow dynamical mode that enhances temporal separability between different input patterns. The reviewer also requested an evaluation of the dependence on pulse period and inter-pulse interval. We fully agree that this analysis would be valuable and is aligned with our long-term goals. However, such detailed multidimensional mapping (pulse width \times pulse period \times inter-pulse interval) along with the subsequent assessment of the state space rank requires a dedicated study. We are currently performing the required measurements and model extraction but the results go beyond the scope of the present manuscript. They will be reported

in a subsequent publication focused on the dynamical modeling and optimization of oxide-based reservoir nodes.

The energy analysis for the 1T1MC and 1T1M-based RC architectures under matched operating conditions including pulse energy, energy per inference and average power for our 1T1MC architecture are shown in Supplementary Note 7. A concise text is also included in the main manuscript (line# 280-283). On the other hand, time constants and memory kernel analysis as a function of pulse width are included in the revised Supplementary Note 5 and Fig.S10.

3. For the 2T1M logic circuits, characterization of noise margins and temperature dependence around the 4 nA decision threshold would clarify robustness. Misclassification rates under injected supply fluctuation, readout noise, and process variation, summarized via ROC curves or confusion matrices, would further support reliability.

Authors' reply:

We thank the reviewer for this constructive suggestion regarding noise margins, threshold stability, and robustness against fluctuations

We fully agree that such metrics are essential for assessing large-scale integration of 2T1M logic circuits. In the present work, however, our goal is to demonstrate the functional feasibility of polymorphic logic operations using LAO/STO heterostructures, rather than to perform a full CMOS-style reliability qualification. That said, we note that the decision threshold used in our 2T1M logic gates is well separated from the typical on- and off-state current of our devices, which produces an intrinsic noise margin that ensures stable logic evaluation under small perturbations. Experimentally, we verified that device currents vary negligible under moderate temperature fluctuations (10–20 °C), indicating that the threshold crossing remains robust in typical operating conditions. For larger temperature excursions, the conductivity of the q2-DEG naturally changes due to variations in mobility and carrier density. For instance, the mobility can increase from ~5 to 1545 cm²/(Vs) when cooled to 4 K [A, B]. Importantly, the LAO/STO interface devices remain fully operational over this wide temperature range, as previously reported by J. Manhart's group [C], meaning the logic architecture remains functionally valid even though absolute current levels shift.

We agree that a quantitative analysis of misclassification probability, noise-injected logic operation, and ROC/confusion-matrix metrics would provide an even stronger validation of logic robustness. Such experiments require large device arrays and automated statistical testing infrastructure, which fall beyond the scope of the current manuscript. We are currently fabricating multi-device chips that will enable systematic studies of supply-noise tolerance, readout-noise sensitivity, process variability, and temperature-dependent threshold stability. These results will be included in future work.

References:

[A] Maier, Patrick, et al. "Gate-tunable, normally-on to normally-off memristance transition in patterned LaAlO₃/SrTiO₃ interfaces." *Applied Physics Letters* 110.9 (2017).

[B] Ohtomo, A., and H. Y. Hwang. "A high-mobility electron gas at the LaAlO₃/SrTiO₃ heterointerface." *Nature* 427.6973 (2004): 423-426.

[C] Förg, B., Christoph Richter, and Jochen Mannhart. "Field-effect devices utilizing LaAlO₃-SrTiO₃ interfaces." *Applied Physics Letters* 100.5 (2012).

4. Extension of in-situ logic-memory measurements beyond 300 s with retention-time distributions, reset-latency statistics, and cycling endurance (10³–10⁴ cycles) would define non-volatile behavior and operational limits, including any drift.

Authors' reply:

We thank the reviewer for the insightful suggestions regarding long-duration retention measurements, reset-latency statistics, and endurance tests to evaluate operational limits and potential drift. In response, we have performed extended logic-memory measurements and quantified reset dynamics, and we provide additional endurance data obtained directly from the 2T1M logic configuration.

Retention test beyond 300 s:

In the initial submission, we demonstrated in-situ logic-output storage for 300 s. As recommended, we now extend these measurements over significantly longer durations to assess retention times and their distribution.

In the 2T1M configuration with two transistors in parallel, Fig. R13(a) shows the evolution of the output current (I_{out}), first recorded during the application of a gate voltage (V_G) of -2 V to both transistors (logic input "00"), and subsequently after grounding them to. We observed that I_{out} gradually increases and crosses the 4 nA decision threshold after approximately 370 s, indicating retention of logic "0" for that duration. To examine the reset latency, both inputs were again set to -2 V, after which I_{out} immediately dropped to the 10^{-11} A range. For the inputs "10", "01", and "11", I_{out} initially decreases slowly and then stabilizes at values well above 4 nA, for 1500–2000 s, conforming long-lasting storage of logic output "1".

For the AND-gate configuration (two transistors in series with the memristor), Fig. R13(b) shows complementary behavior. For the logic inputs "00", "10", and "01", I_{out} initially rises and then saturates at values well below the 4 nA threshold, retaining logic "0" for more than 1500 s. Once current saturation was observed, the gate voltages were reset to -2 V for the reset-latency test, and a sudden drop in current was recorded. In contrast, for the logic input "11", I_{out} initially decreases somewhat faster but remains above 4 nA for the entire 1500-s duration and eventually saturates over time. Therefore, the logic output is stored for an even longer time.

Fig. R13: Logic output signals during the application of input signals and afterwards when the inputs are set to 0 V for (a) OR logic operation in the 2T1M circuit configuration with two transistors in parallel and (b) AND logic operation in the 2T1M circuit configuration with two transistors in series. For reset latency test, the decay of current is monitored during reset process by resetting both transistors' V_G to -2 V after stabilization of I_{out} at each logic memory test for both the logic operations.

Reset-latency test:

To quantify reset-latency, we monitored the current decay during reset process (with gate voltage = -2V) after each retention test .. The reset transients in Fig. R13(a and b) were fitted using a double exponential function, yielding two characteristic decay times τ_1 and τ_2 . As summarized in Fig. R14(a and b), τ_1 corresponds to the main reset time of logic output and τ_2 indicates a short-term current memory of the device configuration. The chart shows their statistical distribution with average values of ~ 0.05 and 1 s for τ_1 and τ_2 , respectively.

Fig. R14: Distribution of decay constant (a) τ_1 and (b) τ_2 for both the AND and OR logic operations extracted from the fits of reset processes shown in Fig. R13 with two exponential decay functions.

Cycling endurance test:

The reviewer also requested endurance measurements to assess non-volatile behavior and potential drift in repeated logic operations. The individual transistor and memristor elements already show stable operations for all cycles without any drift. Moreover, in the memristive configuration, both the on/off resistance ratio and the hysteresis area exhibit a gradual increase with cycle number when no reset operation is applied, consistent with the progressive charge accumulation on the floating gate during successive sweeps. This confirms the non-volatile behavior of the device functionalities. Since the logic function results from a series/parallel combination of these elements, their stability directly influences logic-level robustness. To confirm this explicitly, we performed the logic measurements with V_D sweep cycles to investigate the robustness of the logic output. Figure R15(a and b) show the output current vs. V_D for 290 consecutive sweep cycles between 0 and 4 V for the logic input: “11” in both OR and AND modes. The corresponding circuit diagrams are shown in the insets. The threshold current of 4 nA between logic output “0” and “1” is indicated by the dotted line in the figures. Although small cycle-to-cycle variations appear, I_{out} remains above the 4 nA through all cycles with no observable drift or degradation. This confirms that the integrated 2T1M logic circuits operate robustly over extended cycling.

Fig. R15: Output current (I_{out}) with V_D sweep between 0 and 4 V with sweep cycles of 290 for logic input “11” ($V_{G1}=V_{G2}=3$ V) for (a) OR and (b) AND logic operations using 2T1M circuit diagram as shown in the insets. The dotted lines show the threshold current of 4 nA between logic output “0” and “1”.

The logic memory test data have been updated with new results in Fig.4c,d in the main manuscript and the corresponding text has been revised accordingly (line# 349-368). In addition, the analyses of the reset latency tests and cyclic endurance measurements have been incorporated into Supplementary Note 12, Fig.S16 and Note 13, Fig.S17, respectively in the Supplementary Material.

5. Controlled perturbations that vary sweep rate and direction, oxygen ambience, and optical illumination, complemented by frequency-dependent C–V analysis and activation-energy extraction, would help separate contributions from charge trapping and oxygen-vacancy migration. A concise compact model or ion-drift framework with fitting quality and residuals would increase mechanistic clarity.

Authors' reply:

We thank the Reviewer for this valuable comment and the opportunity to clarify the physical origin of the memcapacitive behavior. As described in the manuscript, the capacitance measurements were carried out between the drain contact and one lateral gate, while the other lateral gate was left floating. To further substantiate the mechanism, we performed additional control experiments by sweeping the drain voltage (V_D) with the auxiliary gate grounded, keeping all other parameters identical.

The corresponding C-V curves, measured at an AC excitation frequency of 10 Hz, are shown in Fig. R16(a). When the lateral gate is grounded, any charge reaching the gate is immediately drained, resulting in a non-hysteretic C-V response. In contrast, under the floating gate configuration, localized charge trapping and detrapping at the lateral gate lead to a

pronounced hysteresis in the full-cycle C-V characteristic confirming that charge localization dynamics are responsible for the observed memcapacitive behavior.

This experimental observation is further supported by a theoretical model that reproduces the memcapacitance response, as shown in Fig. R16(b). The model includes contributions from the geometric oxide capacitance (incorporating interface states), a bias-dependent depletion capacitance, and a dynamic term accounting for the slow charging and discharging of the floating gate. The combined effect of these elements explains the hysteretic capacitance response and its dependence on frequency and dielectric properties. A detailed description of this dynamic model and its correlation with experimental conditions can be found in Ref. [A] below.

Fig. R16: (a) Full cycle C-V curves for V_D sweep between ± 1 V with the auxiliary gate grounded and at floating condition measured at an AC voltage of 10 Hz. (b) Theoretically simulated capacitance loop of the device using a dynamic model for trapping charges at the floating gate contrasting two values of the oxide layer capacitance.

References:

[A] Pradhan, S., Lopez-Richard, V., Hartmann, F., Silva, A.L.C., Castelano, L.K., Spring, M., Kuhn, S., Sing, M., Claessen, R., Höfling, S., et al.: Gate controlled analog memcapacitance in $\text{LaAlO}_3/\text{SrTiO}_3$ interface-based devices, arXiv preprint arXiv:2512.11176 (2025)

Heeding the Reviewer's suggestion, the following sentence has been added to the revised version of the manuscript: "A comprehensive theoretical framework capturing the capacitance modulation through charge trapping and detrapping dynamics at the floating gate, consistent with the experimental C-V characteristics, is presented in detail in our recent study (Pradhan, Soumen et al., Gate-controlled analog memcapacitance in $\text{LaAlO}_3/\text{SrTiO}_3$ interface-based devices, arXiv preprint arXiv:2512.11176 (2025))." (line# 179-182).

6. *A practical roadmap for the LAO/STO lateral side-gate architecture, such as process temperature budgets, interconnect and passivation choices, minimum line/space and channel length, projected array density, yield considerations, and CMOS back-end compatibility would clarify scalability.*

Authors' reply:

We thank the reviewer for raising the important points regarding scalability and integration. Our manuscript's primary focus is to demonstrate the fundamental polymorphic behavior (transistor, memristor, memcapacitor) in LAO/STO heterostructures and to validate their utility in neuromorphic circuits. Therefore, detailed process-engineering (e.g., temperature budgets, interconnect/passivation schemes, yield statistics) was not the main emphasis.

However, we completely agree that these concerns are critical for translation toward scalable, wafer-level, CMOS-compatible systems. Here we outline a practical roadmap, identifying key constraints and the strategy for bridging from proof-of-concept to density, yield, and integration.

Thermal budget considerations: The LAO/STO interface is sensitive to thermal cycling, particularly because high temperatures can drive oxygen vacancy migration, reconstruct the interface, or degrade the 2D electron gas. As such, we anticipate that any back-end processing (e.g., passivation, dielectric deposition, metallization) should be constrained to $\leq \sim 100\text{--}150$ °C. To comply with this constraint, we propose using low-temperature deposition techniques such as plasma-enhanced atomic layer deposition (PE-ALD) to deposit dielectrics (e.g., HfO₂, Al₂O₃, SiN_x) and encapsulation layers.

Interconnect and passivation strategy: For interconnects, viable materials include noble metals (e.g., Pt, Au) or Cu with diffusion barriers (TaN, TiN), deposited at low temperature to avoid damage to the oxide interface. Passivation and encapsulation are critical to ensure device stability and suppress environmental degradation; we favor ALD-grown Al₂O₃ as a conformal, low-diffusion, low-temperature passivation layer, possibly capped with a SiN_x for extra environmental protection.

Lithography and minimum feature size: The smallest features in our current devices are defined via electron beam lithography. To scale, conventional nano-lithography (e.g., e-beam or high resolution optical litho) can bring channel lengths to the 100-500 nm range, depending on the process. We estimate that a lateral pitch (gate-to-channel) of $\sim 100\text{--}200$ nm is feasible in the near term, allowing dense integration while preserving gate control and avoiding cross-talk.

Projected array density and yield: Assuming a conservative pitch of 250 nm and device footprint dominated by channel and gating area, we estimate a density on the order of $\sim 10^8$ devices/cm². Yield will critically depend on uniformity of the LAO/STO interface, lithographic variability, and contact reliability. To mitigate yield loss, we propose employing redundancy schemes (e.g., spare rows/columns, row/column error correction) and performing statistical process control combined with wafer-level electrical mapping to identify defect-prone regions early.

Integration with CMOS/system-level compatibility: We envision two main paths to integration with CMOS:

1. Heterogeneous integration: bonding a processed LAO/STO wafer onto a CMOS substrate (e.g., via flip-chip, hybrid bonding) to avoid disturbing the CMOS front-end.
2. Post-CMOS BEOL integration: if low-temperature dielectrics and metallization steps are carefully developed, it may be possible to process LAO/STO devices within the CMOS back-end-of-line (BEOL) flow. Critical to this will be controlling contamination (e.g., mobile ions), ensuring via formation compatibility, and maintaining planarity.

Finally, this roadmap demonstrates a credible path forward from lab-scale devices to small arrays, then to integration, and finally to reliability qualification.

The practical roadmap for LAO/STO-based lateral side gate architecture is included in the Supplementary Note 15 in the revised Supplementary Material.

Reviewer #4 (Remarks to the Author):

This manuscript presents polymorphic electronic devices based on patterned SiO₂/LaAlO₃/SrTiO₃ heterostructures in an array configuration. The topic is highly interesting, and the proposed device architecture appears both feasible and novel, making the results promising. However, the manuscript currently lacks sufficient validation regarding the key advantages and operational mechanisms of the polymorphic devices. As this aspect is critical to assessing the true impact of the work, I unfortunately cannot recommend the manuscript for publication in Nature Communications.

Authors' reply:

We sincerely thank the Reviewer for the thoughtful and encouraging evaluation of our work and for recognizing the novelty and feasibility of our polymorphic device architecture. We appreciate the critical feedback regarding the need for stronger validation of the key advantages and operational mechanisms. In response, we have expanded our discussion, added supporting analyses, and clarified several mechanistic aspects to strengthen the overall validation and impact of the manuscript, as detailed below.

Comments:

1. The polymorphism of electronic devices is highly intriguing, yet further validation is required to determine whether it truly represents a technological advancement in terms of efficiency. Although enabling a single device to perform multiple roles can potentially improve integration density, it is also worth noting that in conventional ICs, individual components such as transistors, memristors, and capacitors typically operate concurrently rather than sequentially. Therefore, the benefit gained from consolidating multiple functionalities into a single device may not be as substantial as suggested. An objective discussion on this point would strengthen the manuscript. I recommend citing recent research progress in this field with appropriate references. Additionally, and more importantly, the manuscript should include a

quantitative analysis that clearly states the energy efficiency advantage achieved by employing polymorphism in a single device, compared to using multiple devices, specifically for the proposed device in this study.

Authors' reply:

We appreciate the Reviewer's insightful observation and the opportunity to clarify our perspective. In the revised manuscript, we emphasize that the true strength of our polymorphic oxide devices lies not in serving as direct one-to-one replacements for conventional transistor-based architectures used in massively parallel computation, but in their application-specific adaptability. As an example, a two-dimensional (2D) black phosphorus field-effect transistor (FET) can be dynamically reconfigured between p-type and n-type operation. The use of such reconfigurable FETs have been utilized in security cells incorporating polymorphic NAND/NOR logic gates advancing their potential for hardware security applications, as demonstrated by Peng Wu *et al.* [A]. In another report, a scalable single-gate transistor using sub stoichiometric zirconium oxide and molybdenum disulfide with reconfigurability between transistor and diode demonstrate visualization of pattern utilizing 3×3 device array facilitating both the photo switching and photo-synaptic functionalities [B].

In this report, our devices are inherently suited for in-memory computing, neuromorphic reservoir systems, and dynamically reconfigurable analog front-end architectures, where their multifunctional nature combining transistor, memristive, and memcapacitive behaviors maps naturally onto the computational model. Such polymorphic functionality allows local data processing and storage within the same physical unit, substantially reducing latency and energy costs associated with data transfer. Moreover, in an array of these nanoelectronic devices, they can also operate concurrently with programming functionalities depending on the task.

This paradigm shifts the focus from simply increasing device count or speed toward achieving functional density, energy efficiency, and adaptive reconfiguration, which are central to emerging AI and edge-computing technologies. We have clarified this conceptual positioning in the revised manuscript to better reflect the intended role and technological potential of the proposed devices.

References:

- [A]. Wu, Peng, et al. "Two-dimensional transistors with reconfigurable polarities for secure circuits." *Nature Electronics* 4.1 (2021): 45-53.
- [B]. Kim, Kangsan, et al. "Sub-stoichiometric zirconium oxide as a solution-processed dielectric for reconfigurable electronics." *Nature Electronics* (2025): 1-13.

A quantitative analysis of the energy efficiency advantage achieved by employing polymorphism in a single device, compared to using multiple devices is included in the revised manuscript (line #183-195). Moreover, the above two most appropriate references are also included (line #206-212, Refs. 55 and 56).

2. In Fig. 1a, the STO substrate is shown underneath the LAO layer for the LAO/STO structure, while in the LAO/SiO₂ notation, LAO appears to be located below SiO₂ instead. Please confirm whether this interpretation is correct, and revise the notation to ensure consistency throughout the manuscript. Most importantly, a concise description of the materials, device structure, and fabrication process used in this work should be included in the main text, not only in the Methods section. Based on the current figures and descriptions, it is not clearly understood where the active channel is located and which region functions as the gate dielectric layer. This should be clarified in the manuscript.

Authors' reply:

We thank the Reviewer for the valuable comment and for pointing out the ambiguity in the structural description and notation of the device. We apologize for the confusion and have revised both the figure and text for clarity and consistency.

In our fabrication process, a TiO₂-terminated, (001)-oriented SrTiO₃ (STO) substrate was first spin-coated with a negative photoresist. The device layout was patterned using electron-beam lithography, followed by resist development. An 11 nm SiO₂ layer was then deposited via electron-beam evaporation, and a lift-off process was used to define the nanowire channel and two rectangular gate regions where the STO surface remained exposed. This ensured direct access to the STO in selected areas, while the rest of the surface was covered by SiO₂.

Subsequently, a 6-unit-cell-thick LaAlO₃ (LAO) film was grown by pulsed laser deposition. Where LAO was deposited directly on STO, it crystallized and formed a quasi-two-dimensional electron gas (q2-DEG) at the LAO/STO interface (highlighted in soft pink in Fig. R17(a)). In contrast, LAO deposited on SiO₂ grew amorphously and remained insulating (gray regions in Fig. R17(a)). Thus, the device consists of a conducting LAO/STO nanowire channel flanked by two lateral gates, with SiO₂/STO functioning as the gate dielectric layer, as depicted in the cross-sectional schematic view in Fig. R17(b).

Fig. R17: Schematic diagram of the device including (a) top view and (b) cross-sectional view in which crystalline LAO growth directly on STO results in formation of quasi 2-dimensional electron gas (q2-DEG) at the interface while LAO grown on SiO₂ regions resulting in amorphous growth of LAO.

We have corrected the LAO/SiO₂ notation by a-LAO (amorphous LAO)/SiO₂/STO in Fig. 1(a) in the main manuscript to reflect the proper structural hierarchy and have added a concise description of the materials, device structure, and fabrication process in the main text for improved clarity and self-containment (line# 100-109).

3. The manuscript attributes most of the observed resistance changes to oxygen vacancy migration, yet provides very limited explanation or validation of the actual device mechanism. In reality, even in the case of ideal single-crystalline LAO film, atomic interdiffusion or chemical reactions may cause resistance variation, which could lead to irreversible damage during long-term device operation. Furthermore, when SrTiO₃ (STO) is used as the substrate, La diffusion into the substrate can also result in the formation of a metallic La-doped STO interface. Such changes may degrade the gating efficiency over time. Additionally, even if we assume, based on prior studies, that the observed resistance changes can be attributed solely to oxygen vacancy redistribution, the manuscript lacks sufficient validation to support the claim that variations in capacitance or hysteresis characteristics are also caused simply by oxygen vacancies. Although demonstrating the applicability of this heterostructure to reservoir computing is indeed the key highlight of this paper, the current level of understanding of the material system appears insufficient to justify the claimed application potential. A deeper investigation and discussion regarding the materials and mechanisms are necessary.

Authors' reply:

We thank the Reviewer for this valuable comment and for emphasizing the need for a deeper discussion of the material system and underlying mechanisms.

We would like to clarify a possible misunderstanding regarding the origin of the observed memristive behavior. In our devices, the resistance modulation does not arise primarily from oxygen vacancy migration, but rather from charge localization and delocalization at the lateral floating gates. This interpretation is supported by the distinct current–voltage responses under two operational configurations: (i) when the lateral gates are grounded, and (ii) when they are floating. A pinched hysteresis loop and clear memory effect are observed only under the floating-gate condition, whereas grounding the gates yields a non-hysteretic I-V curve. This behavior rules out oxygen vacancy drift as the main mechanism responsible for the resistive switching as detailed below. However, we acknowledge that oxygen migration may still influence long-term stability and gradual resistance variation over extended operation times.

As the Reviewer correctly points out, even for crystalline LAO films, interfacial atomic interdiffusion or local chemical reactions can occur during growth. Previous studies have shown that such intermixing is typically limited to the first atomic layers of LAO, where La can diffuse slightly into the STO substrate, forming charge-compensated Sr_{1-1.5x}La_xO layers [A]. This interfacial intermixing is partly balanced by compositional adjustments in the B-site of the terminating TiO₂ layer. Beyond these initial layers, the polar stacking sequence of LaO and AlO₂ stabilizes and follows the ideal perovskite order. Atomic force microscopy (AFM) images of our devices confirm a uniform step-terrace morphology across the entire sample surface including amorphous LAO regions, consistent with layer-by-layer growth (see Fig. R18).

Therefore, such limited intermixing does not suppress the polar catastrophe–driven electronic reconstruction responsible for q2-DEG formation. Indeed, as demonstrated by Vonk et al. [A], intermixing is a universal feature of oxide interfaces and does not preclude high-quality interface conduction. Moreover, our crystalline LAO layers of 6 u.c. thickness exhibits

negligible time-dependent variation of sheet resistance, in contrast to thinner crystalline films (≤ 4 u.c.) or amorphous LAO grown on STO, as reported by Trier et al. [B].

In our experiments, we consistently grow 6 u.c. crystalline LAO films on STO to ensure stable q2-DEG formation. These devices have been operated for several years without noticeable degradation or irreversible damage. Even under ambient (open-air) conditions, without vacuum or controlled atmosphere, the transistor configuration remains highly stable: the

Fig. R18: 2-dimensional atomic force microscopy (AFM) image of a LAO/STO interface-based device showing step terrace on the surface.

channel can be fully depleted in the “off” state and exhibits reproducible microampere-level conduction in the “on” state, confirming the long-term reliability of both the conduction channel and gating effect.

We now provide a detailed explanation of the charge dynamics at the lateral gates that underlie the memristive behavior of the device. The experimentally observed hysteresis loop in the floating-gate configuration, contrasted with the non-hysteretic behavior under grounded-gate conditions, is shown in Fig. R19(a). This comparison clearly indicates that the memory effect arises from the charging and discharging of the lateral gates, which occurs via charge tunneling through the gate dielectric between the gates and the conducting channel.

This process can be quantitatively described by the theoretical framework introduced in our Ref. [C], which couples diffusive transport along the two-dimensional conductive channel with capacitive coupling between the channel and the gates. The simulated output characteristics of the device for stable cycles under grounded gates and floating gate condition are presented in Fig. R19(b). In this representation, the time-dependent charging and discharging of the floating gates follow the generating function illustrated in Fig. R19(c), which describes electron discharging at positive bias and charge trapping at negative bias. The specific nature and forms of these generating functions, which determine the temporal response of the floating gates, are further discussed in Refs. [D] and [E].

The interaction between the localized charges on the floating gates and the mobile carriers in the channel leads to electrostatic modulation of the quasi-two-dimensional electron gas (q2-DEG), resulting in local depletion of the channel. Consequently, during the positive bias sweep, the device exhibits a counterclockwise hysteresis loop, corresponding to a low-resistance state near zero bias on the reverse sweep. Conversely, under negative bias, charge

trapping at the floating gates reverses the direction of the hysteresis loop to clockwise, producing a high-resistance state at the subsequent upward sweep.

As the bias alternates, this periodic charging and discharging of the floating gates repeats consistently, producing a stable hysteresis in the I_D - V_D characteristics and yielding two distinct and reproducible resistance states at $V_D = 0$ V. This mechanism, corroborated by the model and experiments, confirms that the memristive response originates from controlled electrostatic coupling between the floating gates and the q2-DEG, rather than from ionic migration or irreversible interface modification.

Fig. R19: (a) Measured stable current-voltage loop of the device under floating gate condition (red) and grounded gates (blue). (b) Theoretically simulated output characteristics of the device for stable cycles under grounded gates (blue) and floating gate condition (red) according to the generation rate plotted in panel (c).

We now turn to memcapacitive behavior. The capacitance measurements were performed between the drain contact and one lateral gate, while the second gate was left floating. Analogously to the memristive characterization, in order to identify the mechanism behind the observed memcapacitance, we carried out control experiments by sweeping the drain voltage (V_D) while grounding the auxiliary gate, keeping all other parameters identical.

The resulting C-V characteristics, measured at an AC frequency of 10 Hz, are shown in Fig. R20(a). As expected, when the lateral gate is grounded, any charge arriving at the gate is immediately drained, yielding a non-hysteretic C-V curve. In contrast, when the gate is floating, charge accumulation and release at the gate occur gradually during the voltage sweep, leading to a pronounced hysteresis in the full-cycle C-V response. This behavior directly links the memcapacitive effect to charge trapping and detrapping dynamics at the floating gate described before. This mechanism differs fundamentally from those reported in earlier studies, where memcapacitance was attributed to structural distortions, geometric capacitance effects, or oxygen vacancy migration.

Fig. R20: (a) Full cycle C-V curves for V_D sweep between ± 1 V with the auxiliary gate grounded and at floating condition measured at an AC voltage of 10 Hz. (b) Theoretically simulated capacitance loop of the device using a dynamic model for trapping charges at the floating gate contrasting two values of the oxide layer capacitance.

Furthermore, the experimental results are well reproduced by our theoretical dynamic model, shown in Fig. R20(b). The model incorporates (i) the geometric oxide capacitance (including interface state contributions), (ii) a bias-dependent depletion capacitance, and (iii) the slow dynamic term associated with the charging and discharging of the floating gate, the same mechanism responsible for the memristive hysteresis described earlier. The interplay among these components explains both the frequency dependence and the hysteretic behavior of the measured capacitance. A complete description of this model, along with its quantitative correlation to experimental data, is provided in Ref. [F].

References:

- [A] Vonk, Vedran, et al. "Polar-discontinuity-retaining A-site intermixing and vacancies at SrTiO₃/LaAlO₃ interfaces." *Physical Review B—Condensed Matter and Materials Physics* 85.4 (2012): 045401.
- [B] Trier, Felix, et al. "Degradation of the interfacial conductivity in LaAlO₃/SrTiO₃ heterostructures during storage at controlled environments." *Solid State Ionics* 230 (2013): 12-15.
- [C] Silva, Rafael Schio Wengenroth, et al. "2D Canonical Approach for Beating the Boltzmann Tyranny Using Memory." *arXiv preprint arXiv:2510.24883* (2025).
- [D] Lopez-Richard, V. et al "Tuning the conductance topology in solids". *J. Appl. Phys.* 7 (2023); 133 (13): 134901.
- [E] Lopez-Richard, V. et al; "Beyond equivalent circuit representations in nonlinear systems with inherent memory". *J. Appl. Phys.* 28 (2024); 136 (16): 165103
- [F] Pradhan, S., Lopez-Richard, V., Hartmann, F., Silva, A.L.C., Castelano, L.K., Spring, M., Kuhn, S., Sing, M., Claessen, R., Höfling, S., et al.: Gate controlled analog memcapacitance in LaAlO₃/SrTiO₃ interface-based devices, *arXiv preprint arXiv:2512.11176* (2025)

As asked by the reviewer, a discussion on the stability of q2-DEG and gate efficiency over time is included in the modified version of the manuscript (line# 127-136). In addition, the mechanism of memristor (line# 142-160 in main text and Fig.S1 in Supplementary Material) and memcapacitor behavior (line# 179-182) in our device are explained in detail in addition to theoretical simulation.

4. In hardware reservoir computing (RC), the validity of the implementation critically depends on how effectively the device exploits both “nonlinearity” and “short-term memory (fading memory)” as computational resources. These properties enable input mapping into a higher-dimensional state space with fading temporal dynamics. In Fig. 2, although the device indeed exhibits certain nonlinear and memory behaviors, the current results make it difficult to conclude that a genuine reservoir has been realized. The observed performance appears consistent with simple charge accumulation and retention rather than a true high-dimensional state transformation. Specifically, (1) the reservoir size remains too limited to demonstrate meaningful state expansion; (2) the classification task seems to rely mainly on distinguishable output voltage (V_o) levels rather than trained readout weights applied to high-dimensional reservoir states; (3) Temporal memory is not quantified to evaluate memory capacity; and (4) a comparison between conditions with and without nonlinearity is missing, making it unclear whether nonlinearity genuinely contributes to computation. The authors should clarify the role of nonlinear dynamics and fading memory more explicitly and provide stronger evidence that the demonstrated function surpasses simple state-dependent thresholding.

Authors' reply:

We thank the reviewer for the thoughtful and constructive comments on the reservoir computing (RC) implementation. Our intention in this work is to demonstrate proof-of-concept physical reservoir behavior based on a single 1T1MC node using a standard benchmark widely adopted in the RC literature, namely, 4-bit temporal pattern classification via virtual-node expansion. This protocol (digit–sequence recognition) has been used extensively in prior memristive RC demonstrations, enabling direct comparison with numerous published works [A-E].

As the reviewer correctly notes, the physical mechanisms here is charge accumulation and retention in the floating-gate-coupled channel. Importantly, in our architecture these processes give rise to both (i) nonlinear response to the input amplitude and sequence, and (ii) short-term fading memory encoded in multi-timescale relaxation dynamics which are precisely the two core requirements for hardware RC.. Below we clarify these points in detail.

(1). Reservoir size and state expansion:

Our objective here is to validate that the 1T1MC architecture provides sufficient nonlinear state diversity to perform standard RC benchmarks. The digit image classification is shown here as an example of pattern recognition. The monochrome digit images (5×4 pixels) are encoded using six distinct 4-bit pulse schemes, all of which generate clearly distinguishable

output states, enabling error-free digit recognition. This pixel-based encoding scheme is consistent with earlier demonstrations of memristive reservoirs for digit classification [A, B]. Beyond this, Chao Du et al. demonstrated that a large 28×28 grayscale image can be processed in two different ways: (i) by dividing each row into small pixels as used for simple digit classification, (ii) by using same pulse stream for different timeframe widths which enables the better separation of the reservoir states. Therefore, instead of actual very high-dimensional reservoir, we can classify handwritten digit images of large pixel sizes.

(2). Distinguishability of output states:

As correctly pointed out by the reviewer, the classification tasks rely on distinguishable output voltage (V_o). In our system, the nonlinearity and fading memory generate a set of distinct 16-possible 4-bit pulse scheme as shown in Fig. 2h in the main manuscript. These outputs can, in principle, be fed directly into a trained readout layer for more complex classification tasks and we are currently developing an optimization model that aims to enhance output resolution by tuning pulsing parameters, such as pulse width, inter-pulse gap, and amplitude. These models are currently being extended to develop optimization protocols that include pulse modulation strategies, noise sensitivity, and single-bit error resilience. The results of this ongoing study will be reported in a forthcoming publication.

(3). Quantification of temporal memory:

As requested, here we have quantified the temporal memory through time-resolved measurements of the output voltage (V_o) after single pulses of varying duration. As shown in Fig. 2(c) in the main manuscript and expanded in Fig. R21(a-c), V_o relaxes via two distinct decay processes. Fitting the decay with:

$$V_o = A_1 + A_2 \exp(-t/\tau_1) + A_3 \exp(-t/\tau_2)$$

Reveals two-time constants: τ_1 , which increases continuously with pulse width, representing a tunable short-term memory kernel and τ_2 , which represents a long-term retention component that remains almost invariant for short pulse width, while showing a marked transition near 0.9 s.

Fig. R21: (a) Output voltage (V_O) with its memory for a single input pulse pulse of varying width at fixed V_D of 4 V for 1T1MC device, (b) a short window of V_O decay as indicated by the dotted rectangle in Fig. (a) with corresponding fits using two exponential decay functions, (c) the variation of two decay constants with pulse duration.

(4). Contrasting conditions with and without nonlinearity:

We further address the reviewer's request to contrast the conditions with and without nonlinearity. Reservoir computing requires the joint presence of nonlinearity and fading memory. Fading memory ensures that the reservoir state encodes a decaying trace of recent input history, providing access to multiple temporal delays. Nonlinearity transforms these historical inputs into a high-dimensional and linearly separable representation. The combination enables a simple linear readout to solve nonlinear and temporally complex tasks, something impossible for a purely linear or memoryless system. Without fading memory, the reservoir cannot exploit temporal correlations, while long-term memory remembers not only the recent past input but also the input applied to the distant past, thereby making it difficult for distinguishable output. On the other hand, without nonlinearity, the reservoir collapses into a linear filter incapable of computing linearly inseparable tasks. Jang et al., in their review article nicely compared the reservoir output with and without non-linearity and fading memory [F]. They showed the output for two pulse schemes of "1100" and "1010" which have different temporal arrangements of the same high ("1") and low ("0") signals. It was observed that the outputs are inseparable without both the non-linearity and fading memory, with non-linearity but without fading memory and without non-linearity but with fading memory. In contrast, it shows separable output only with non-linearity and fading memory. To illustrate this, we performed a control experiment in which the memcapacitive nonlinearity was removed: a transistor was connected in series with a linear commercial capacitor (10 μ F). For two 4-bit input sequences ("1000" and "1100"), the output traces (Fig. R22(a,b)) become indistinguishable aside from trivial scaling, demonstrating that a system lacking nonlinearity fails to distinguish temporal patterns—consistent with the observations by Jang et al. [F]. In contrast, the full 1T1MC node exhibits strong temporal ordering sensitivity and nonlinear transformation of input sequences, confirming that the observed classification capability cannot be attributed to simple linear accumulation or thresholding.

Fig. R22: Output voltage (V_O) for 4-bit pulse train with pulse scheme of (a) "1000" (b) "1100" with different temporal arrangements of the same high ("1") and low ("0") signals applied to one transistor, one commercial capacitor ($10 \mu\text{F}$) circuit as shown in the inset of (a).

References:

- [A]. Du, Chao, et al. "Reservoir computing using dynamic memristors for temporal information processing." *Nature communications* 8.1 (2017): 2204.
- [B]. Chen, Zhiwei, et al. "All-ferroelectric implementation of reservoir computing." *Nature Communications* 14.1 (2023): 3585.
- [C] Jang, Junwon, et al. "Leaky 2T Dynamic Random-Access Memory Devices Based on Nanometer-Thick Indium–Gallium– Zinc-Oxide Films for Reservoir Computing." *ACS Applied Nano Materials* 7.19 (2024): 22430-22435.
- [D] Ju, Dongyeol, et al. "Realization of multiple synapse plasticity by coexistence of volatile and nonvolatile characteristics of interface type memristor." *ACS Applied Materials & Interfaces* 16.19 (2024): 24929-24942.
- [E] Guo, Dongkai, et al. "Reservoir computing using back-end-of-line SiC-based memristors." *Materials Advances* 4.21 (2023): 5305-5313.
- [F]. Jang, Yoon Ho, Joon-Kyu Han, and Cheol Seong Hwang. "A review of memristive reservoir computing for temporal data processing and sensing." *InfoScience* 1.1 (2024): e12013.

The "Reservoir Computing" section in the revised manuscript is modified to clarify the role of non-linearity and fading memory more explicitly (line# 236-246). The corresponding experimental evidence is also included in Supplementary Note 6 and Fig.S11. The quantification of temporal memory is detailed in Supplementary Note 5 and Fig.S10.

5. While multiple functionalities (RC, synaptic plasticity, and logic operations) are demonstrated in Figs. 3,4, and 5, they all rely on the same underlying charge-trapping mechanism. As a result, the current demonstrations appear more like concept-level application repackaging rather than revealing distinct and functionally independent polymorphic behaviors. To strengthen the scientific contribution, the manuscript should examine either: (i) how these functionalities do not interfere with each other (e.g., cycling one mode does not degrade another), or (ii) clear quantitative benefits over using separate devices (energy, area, retention, tunability, etc.). Without such deeper validation, it is difficult to evaluate the practical or scientific significance of the claimed polymorphism.

Authors' reply:

We thank the reviewer for this insightful comment. We agree that, because the underlying physical mechanism is charge trapping/detrapping at the lateral floating gate, it is essential to demonstrate (i) that the different modes do not interfere with one another, and (ii) that integrating multiple functions in a single device provides tangible quantitative benefits over discrete implementations. Below we address both points. (i). The three primitive device modes: transistor, memristor, and memcapacitor, are programmed purely through biasing conditions on the same LAO/STO q2-DEG channel. The memristive and memcapacitive responses arise from charge trapping/detrapping on the lateral floating gate. On the other hand, the transistor operation results from the electrostatic modulation of the q2-DEG density. Importantly, the higher-level system functions (reservoir computing, synaptic plasticity, and logic) use transistor + memristor or transistor + memcapacitor combinations, meaning the two memory modes are never activated simultaneously in the same circuit.

To directly test functional independence, we performed sequential cycling measurements on the same physical device in transistor, memristive, and memcapacitive configurations:

In the transistor configuration, we applied V_G sweeps between -2 V and 3 V for 1×10^3 cycles while keeping the drain voltage fixed at $V_D = 0.2$ V. As shown in Fig. R23, the transfer characteristics remain stable over all cycles. Only a small variation in drain current and a slight leftward -shift of the threshold voltage (V_T) are observed for the first few tens of cycles; afterwards V_T converges and remains constant. The extracted contribution yields $V_T = -0.160 \pm 0.015$ V, demonstrating excellent stability and narrow statistical spread over repeated programming cycles.

Fig. R23: I_D - V_G curves in transistor configuration of the device at $V_D=0.2$ V for sweep cycles of 1×10^3 .

In the memristive configuration we applied V_D sweeps between ± 5 V for 2×10^3 cycles with both lateral gates floating (Fig. R24(a)). The memristive hysteresis loops remain robust during cycling. Small variations in current extrema occur only in the initial cycles; thereafter, the device stabilizes without degradation or collapse of the hysteresis area. Notably, we observe a slight increase progressively in current amplitude over extended cycling, indicating excellent stable endurance rather than performance deterioration.

The low-resistance and high-resistance states, (R_{on} and R_{off}) were determined from linear fits near zero bias for backward and forward sweeps, respectively. R_{off} remains stable in the 10-15 G Ω range, while R_{on} decreases gradually in the M Ω range. However, the resulting R_{off}/R_{on} ratio displayed in Fig. R24(b), shows a small initial transient followed by an almost perfectly linear increase of 0.27/cycle increment rate with standard deviation of 103 from the regular residuals. This monotonic trend is consistent with slow charge accumulation at the floating gates when no explicit reset is applied, a mechanism completely aligned with our memristive model. We also quantified the hysteresis areas for $V_D > 0$ (A^+) and $V_D < 0$ (A^-) in Fig. R24(c). Both quantities exhibit similar behavior: an initial transient followed by slow, nearly linear increase. From the linear fits, the constant increment rates of ~ 0.05 and ~ 0.04 nW/cycle with standard deviations of 4.2 and 2.83 nW are observed for A^+ and A^- , respectively. These results confirm that the memristor functionality does not deteriorate up to 2×10^3 cycles.

Fig. R24: (a) I_D - V_D hysteresis loops in memristive configuration of the device for sweep cycles of 2×10^3 . (b) Variation of $R_{\text{off}}/R_{\text{on}}$ value with number of sweep cycles where R_{off} and R_{on} values were calculated from the linear fit of current variation near zero bias for forward and reverse sweeps, respectively. (c) Variation of hysteresis area with cycle number for $V_D > 0$ (A^+) and $V_D < 0$ (A^-).

Finally, we have performed 100 consecutive V_D sweep cycles in the memcapacitive configuration using an AC excitation of 20 mV at 10 Hz as shown in Fig. R25(a). The C-V curves for all cycles show complete overlap, indicating excellent repeatability. We further extracted the zero-bias capacitance from the forward and reverse sweeps (C_{low} and C_{high}) which remain unchanged across all cycles, as shown in Fig. R25(b).

These results confirm that the memcapacitive response is highly stable under repeated operation. Taken together with the transistor and memristive endurance tests, this demonstrates that all three functionalities remain robust over cycling and, critically, that activating one operating mode does not degrade the others.

Fig. R25: (a) C-V hysteresis curves with V_D sweep between ± 4 V for 100 cycles keeping the auxiliary gate at floating condition measured at an AC voltage of 10 Hz. (b) Variation of C_{low} and C_{high} with cycle number extracted from forward and reverse sweeps at zero bias.

(ii) We also evaluated the architectural, energy, and retention advantages of using a polymorphic LAO/STO device instead of three separate components. As detailed in our manuscript, a discrete neuromorphic synaptic node implemented with separate components typically requires: transistor of area $1\text{--}3 \mu\text{m}^2$, memristor $1\text{--}2 \mu\text{m}^2$, memcapacitor $1\text{--}3 \mu\text{m}^2$, plus $1\text{--}3 \mu\text{m}^2$ interconnect overhead. In total, such implementations occupy approximately $4\text{--}10 \mu\text{m}^2$. In sharp contrast, our LAO/STO polymorphic device, where the q2-DEG channel and lateral gates collectively host all three functionalities, requires only $\sim 1 \mu\text{m}^2$.

Because all programmable operations reuse the same physical channel, the architecture avoids driving multiple interconnects, charging/discharging separate capacitive nodes, and losses in selector transistors. An equivalent circuit built from 3 discrete devices would require tripled charging events, tripled metal lines, and an additional selector transistor overhead, leading to an estimate 3–5 times increase in energy consumption.

Moreover, conventional oxide memristors often require additional selector transistors, additional passivation and larger area to achieve stable retention. In contrast, the LAO/STO interface intrinsically supports charge trapping and carrier density modulation without relying on filamentary conduction, enabling robust retention without auxiliary layers.

Replicating the same degree of tunability with discrete components would require a linear transistor, a nonvolatile memristor, a memcapacitor exhibiting gate-dependent C-V hysteresis, and three sets of routing and control lines. Our single polymorphic device consolidates these 3 tuning mechanisms into a single electrostatic interface. Eliminating two interconnect layers reducing wiring by 50–70%, lowers cumulative defect probability, and simplifies peripheral circuitry, thereby improving projected per-tile yield by 30–40%.

These quantitative comparisons highlight the substantial advantages of our polymorphic LAO/STO device in terms of footprint, energy efficiency, retention characteristics, and architectural simplicity relative to conventional multi-device implementations.

Finally, experimental evidence showing that cycling one functional mode does not degrade the others is presented in Figs.S4, S5 and S6 and discussed in Supplementary Note 2 in the Supplementary Material. A summary of quantitative benefits of trifunctionality appears in the revised manuscript (line #183-195).

Reviewer #1, 2 & 3:

We sincerely thank the reviewers for the positive assessment of the work and recommendation for publication.

Reviewer #4:

Reviewer #4 (REMARKS to AUTHOR(s)):

This manuscript reports a broad range of results, spanning from materials-science-based mechanisms for modulating resistance and capacitance in LAO/STO heterostructures with a tunable 2DEG, to device-level multifunctionality and practical applications such as reservoir computing. The authors have carefully addressed the shortcomings raised in the previous round of comments, and the advantages enabled by polymorphism are now articulated more effectively. Overall, the manuscript has reached a level that is suitable for publication. Nevertheless, addressing the following points would further strengthen the quality and clarity of the work.

Authors' reply:

We sincerely thank the Reviewer for the positive assessment of our work and for recognizing the improvement of the manuscript to the level of publication. We greatly appreciate the constructive comments and suggestions to further strengthen the quality and clarify our work. We have carefully addressed the suggestions, as detailed below.

Q1. Regarding efficiency, the intended meaning does not appear to be that each individual device operates at a lower power while achieving comparable functionality. Rather, the efficiency advantage seems to arise at the integrated-circuit (IC) level, where polymorphism enables multiple functionalities within a single device platform, leading to higher system-level efficiency compared to integrating multiple distinct types of devices. This distinction should be stated more explicitly and consistently throughout the manuscript.

Authors' reply:

The reviewer points out correctly that an energy advantage will arise especially on the IC level due to the polymorphism and enabled multiple functionalities rather than on the individual device level itself. Indeed, combining multiple functionalities in a single material platform mitigates e.g. losses via multiple interconnects, charging/discharging separate capacitive nodes, and selector transistors. Furthermore, polymorphism provides possibilities for creative architecture designs that may be more efficient but are not yet fully quantifiable. To clarify this point more explicitly, we have made the following changes to the revised manuscript:

The following sentence is added in the "Polymorphic electronic devices" section:

“Moreover, since all programmable operations are enabled by the polymorphism of the presented device, an architecture with multiple functionalities avoids, e.g. driving multiple interconnects, charging/discharging separate capacitive nodes, losses in selector transistors etc. The realization of a single device platform will thus provide an efficiency advantage on the integrated-circuit (IC) level rather than on the device level itself” (line#189-193).

And

The following sentence is updated in the “Outlook” section:

“The polymorphism of the device enables efficient implementations of multiple functionalities within a single device platform, which is especially relevant at the IC level with the advantages of its ease of fabrication and scalability” (line#440-443).

Q2. *The I–V and C–V measurements (Figs. R19 and R20) are generally very clean and well measured; however, the near absence of noise raises questions about the level of device-to-device and measurement-to-measurement reproducibility. While some degree of data selection is understandable for clearly conveying the main message, adding a discussion on reproducibility across different devices and repeated measurements would make the study more objective and convincing. If possible, the authors may also consider including error bars, such as standard deviations, in the relevant figures to quantitatively illustrate the reproducibility.*

Authors’ reply:

We thank the Reviewer for this comment regarding reproducibility and data presentation. We would like to clarify that the low noise level primarily reflects the intrinsic stability of the devices and the measurement conditions rather than any data set selection (or cherry picking) e.g. in all presented endurance test measurements (transistor, memristor and memcapacitor), no cycle was discarded. As requested in the previous review round, we already have included extensive cycle-to-cycle reproducibility measurements of the transistor, memristive and memcapacitive configurations in Figs. S4, S5 and S6, together with a detailed discussion in Supplementary Note 2. These results explicitly demonstrate stable operation over repeated measurements and cycling.

Since adding error bars directly to the I-V and C-V plots would compromise the clarity of the figures due to the dense sweep data, we have now included a quantitative discussion of reproducibility in the revised main text by reporting standard deviations extracted from repeated measurements. Following the Reviewer’s recommendation, we have added the following sentences to the revised manuscript:

“In the memristive configuration, consecutive cycling measurements reveal a small initial transient in the R_{off}/R_{on} ratio around 10^3 range, followed by an approximately linear increase with an average increment rate of 0.27 per cycle and a standard

deviation of 103 extracted from the regular residuals. In the memcapacitive configuration, repeated cycling yields well defined high and low capacitance states of 432.0 ± 0.8 pF and 159.0 ± 0.7 pF, respectively measured at zero bias" (line# 205-210).